# THE IMPACT OF APPROXIMATION ERRORS ON WARM-START REINFORCEMENT LEARNING: A FINITE-TIME ANALYSIS

## ABSTRACT

Warm-Start reinforcement learning (RL), aided by a prior policy obtained from offline training, is emerging as a promising RL approach for practical applications. Recent empirical studies have demonstrated that the performance of Warm-Start RL can be improved *quickly* in some cases but become *stagnant* in other cases, calling for a fundamental understanding, especially when the function approximation is used. To fill this void, we take a finite time analysis approach to quantify the impact of approximation errors on the learning performance of Warm-Start RL. Specifically, we consider the widely used Actor-Critic (A-C) method with a prior policy. We first quantify the approximation errors in the Actor update and the Critic update, respectively. Next, we cast the Warm-Start A-C algorithm as Newton's method with perturbation, and study the impact of the approximation errors on the finite-time learning performance with inaccurate Actor/Critic updates. Under some general technical conditions, we obtain lower bounds on the sub-optimality gap of the Warm-Start A-C algorithm to quantify the impact of the bias and error propagation. We also derive the upper bounds, which provide insights on achieving the desired finite-learning performance in the Warm-Start A-C algorithm.

## 1 INTRODUCTION

Online reinforcement learning (RL) (Kaelbling et al., 1996; Sutton & Barto, 2018) often faces the formidable challenge of high sample complexity and intensive computational cost (Kumar et al., 2020; Xie et al., 2021), which hinders its applicability in real-world tasks. Indeed, this is the case in portfolio management (Choi et al., 2009), vehicles control (Wu et al., 2017; Shalev-Shwartz et al., 2016) and other time-sensitive settings (Li, 2017; García & Fernández, 2015). To tackle this challenge, Warm-Start RL has recently garnered much attention (Nair et al., 2020; Gelly & Silver, 2007; Uchendu et al., 2022), by enabling online policy adaptation from an initial policy pre-trained using offline data (e.g., via behavior cloning or offline RL). One main insight of Warm-Start RL is that online learning can be significantly accelerated, thanks to the bootstrapping by an initial policy.

Despite the encouraging empirical successes (Silver et al., 2017; 2018; Uchendu et al., 2022), a fundamental understanding of the learning performance of Warm-Start RL is lacking, especially in the practical settings with function approximation by neural networks. In this work, we focus on the widely used Actor-Critic (A-C) method (Grondman et al., 2012; Peters & Schaal, 2008), which combines the merits of both policy iteration and value iteration approaches (Sutton & Barto, 2018) and has great potential for RL applications (Uchendu et al., 2022). Notably, in the framework of abstract dynamic programming (ADP) (Bertsekas, 2022a), the policy iteration method (Sutton et al., 1999) has been studied extensively, for warm-start learning under the assumption of accurate updates. In such a setting, policy iteration can be regarded as a second-order method in convex optimization (Grand-Clément, 2021) from the perspective of ADP, and can achieve *super-linear* convergence rate (Santos & Rust, 2004; Puterman & Brumelle, 1979; Boyd et al., 2004). Nevertheless, when the A-C method is implemented in practical applications, the approximation errors are inevitable in the Actor/Critic updates due to many implementation issues, including function approximation using neural networks, the finite sample size, and the finite number of gradient iterations. Moreover, the error propagation from iteration to iteration may exacerbate the 'slowing down' of the convergence and have intricate impact therein. Clearly, the (stochastic) accumulated errors may throttle the

convergence rate significantly and degrade the learning performance dramatically (Fujimoto et al., 2018; Uehara et al., 2021; Dalal et al., 2020; Doan et al., 2019). Thus, it is of great importance to characterize the learning performance of Warm-Start RL in practical scenarios; and the primary objective of this study is to take steps to build a fundamental understanding of the impact of the approximation errors on the finite-time sub-optimality gap for the Warm-Start A-C algorithm, i.e.,

*Whether and under what conditions online learning (e.g., A-C) can be significantly accelerated by a warm-start policy from offline RL?*

To this end, we address the question in two steps: **(1) We first focus on the characterization of the approximation errors via finite time analysis, based on which we quantify its impact on the sub-optimality gap of the A-C algorithm in Warm-Start RL.** In particular, we analyze the A-C algorithm in a more realistic setting where the samples are Markovian in the rollout trajectories for the Critic update (different from the widely used i.i.d. assumption). Further, we consider that the Actor update and the Critic update take place on the *single-time scale*, indicating that the time-scale decomposition is not applicable to the finite-time analysis here. We tackle these challenges using recent advances on Bernstein's Inequality for Markovian samples (Jiang et al., 2018; Fan et al., 2021b). By delving into the coupling due to the interleaved updates of the Actor and the Critic, we provide upper bounds on the approximation errors in the Critic update and the Actor update of online exploration, respectively, from which we pinpoint the root causes of the approximation errors.

**(2) We analyze the impact of the approximation errors on the finite-time learning performance of Warm-Start A-C.** Based on the approximation error characterization, we treat the Warm-Start A-C algorithm as Newton's method with perturbation, and study the impact of the approximation errors on the finite-time learning performance of Warm-Start A-C. For the case when the approximation errors are biased, we derive lower bounds on the sub-optimality gap, which reveals that even with a sufficiently good warm-start, the performance gap of online policy adaptation to the optimal policy is still bounded away from zero when the biases are not negligible. Further, we also derive the upper bounds, which shed light on designing Warm-Start A-C to achieve desired finite-time learning performance. We present the experiments results to further elucidate our findings in Appendix K.

**Related Work. (Warm-Start RL)** AlphaZero (Silver et al., 2017) is one of the most remarkable successes in Warm-Start RL. In a line of very recent works (Gupta et al., 2020)(Ijspeert et al., 2002)(Kim et al., 2013) on Warm-Start RL, the policy is initialized via behavior cloning from offline data and then is fine-tuned with online reinforcement learning. A variant of this scheme is proposed in Advanced Weighted Actor Critic (Nair et al., 2020) which enables quick learning of skills across a suite of benchmark tasks. In the same spirit, Offline-Online Ensemble (Lee et al., 2022) leverages multiple Q-functions trained pessimistically offline as the initial function approximation for online learning. Jump-start RL (Uchendu et al., 2022) utilizes a guided-policy to initialize online RL in the early phase with a separate online exploration-policy. The guided-policy will be abandoned as the online exploration-policy improves. However, a fundamental characterization of the finite-time performance of Warm-Start RL is still lacking. Recent work (Xie et al., 2021) provides a quantitative understanding on the policy fine-tuning problem in episodic Markov Decision Processes (MDPs) and establishes the lower bound for the sample complexity, where no function approximation is used. Our work aims to take steps to quantify the impact of approximation error on online RL when a warm-start policy is given.

**(Actor-Critic as Newton's Method)** The intrinsic connection between the A-C method and Newton's method can be traced back to the convergence analysis of policy iteration in MDPs with continuous action spaces (Puterman & Brumelle, 1979). The connection is further examined later in a special MDP with discretized continuous state space (Santos & Rust, 2004). Recent work (Bertsekas, 2022b) points out that the success of Warm-Start RL, e.g., AlphaZero, can be attributed to the equivalence between policy iteration and Newton's method in the ADP framework, which leads to the superlinear convergence rate for online policy adaptation. Under the generalized differentiable assumption, it has also been proved theoretically that policy iteration is the instances of semi-smooth Newton-type methods to solve the Bellman equation (Gargiani et al., 2022). While some prior works (Grand-Clément, 2021) have provided theoretical investigation of the connections between policy iteration and Newton's Method, the studies are carried out in the abstract dynamic programming (ADP) framework, assuming accurate updates in iterations. Departing from the ADP framework, this work treats the A-C algorithm as Newton's method in the presence of approximation errors, and focuses on the finite-time learning performance of Warm-Start RL.

**(Finite-time analysis for Actor-Critic methods)** Among the existing works on the finite time analysis of A-C methods with function approximation, (Yang et al., 2019) establishes the global convergence under the linear quadratic regulator. (Wang et al., 2020) proves the convergence behavior when both Actor and Critic are approximated by overparameterized neural networks. (Kumar et al., 2019) considers the sample complexity under i.i.d. assumptions where the Actor update and Critic update can be 'decoupled'. Khodadadian et al. (2022) considers the two-timescale setting with Markovian samples. (Fu et al., 2020) focuses on the more general single-time scale setting but constrains the policy function approximation in the energy based function class.

## 2 BACKGROUND

**Markov Decision Processes.** We consider a MDP defined by a tuple $(\mathcal{S}, \mathcal{A}, P, r, \gamma)$, where $\mathcal{S} = \{1, 2, \cdots, n\}$, $n < \infty$ and $\mathcal{A} = \{1, 2, \cdots, A\}$, $A < \infty$ represent the finite state space and finite action space, respectively. $P(s'|s, a) : \mathcal{S} \times \mathcal{A} \times \mathcal{S} \to [0, 1]$ is the probability of the transition from state $s$ to state $s'$ by applying action $a$ and $r(s, a) : \mathcal{S} \times \mathcal{A} \to \mathbb{R}$ is the corresponding reward. $\gamma \in (0, 1)$ is the discount factor. At each step $t$, an agent moves from the current state $s_t$ to next state $s_{t+1}$ by taking an action $a_t$ following the policy $\pi : \mathcal{S} \to \mathcal{A}$ and receives the reward $r_t$. In the Warm-Start RL, we assume that the initial policy $\pi_0$ is given, e.g., in the form of a neural network (Li, 2017), and obtained by offline training. For brevity, we use bold symbols $\boldsymbol{r}_\pi \in \mathbb{R}^n : [r_\pi]_s = r(s, \pi(s))$ and $\boldsymbol{P}_\pi \in \mathbb{R}^{n \times n} : [P^\pi]_{s,s'} \triangleq P(s'|s, \pi(s))$ to denote the reward vector and the transition matrix induced by policy $\pi$. We further denote by $d^\pi : \mathcal{S} \to [0, 1]$ and $\rho^\pi : \mathcal{S} \times \mathcal{A} \to [0, 1]$ the stationary state distribution and state-action transition distribution induced by policy $\pi$. We use $\rho_0$ to represent the initial state distribution. We use $\| \cdot \|$ or $\| \cdot \|_2$ to represent the 2-norm.

**Value Functions.** For any policy $\pi$, define the value function $v^\pi(s) : \mathcal{S} \to \mathbb{R}$ as $v^\pi(s) = \mathbf{E}_{a_t \sim \pi(\cdot|s_t), s_{t+1} \sim P(\cdot|s_t, a_t)} \left[ \sum_{t=0}^\infty \gamma^t r_t | s_0 = s \right]$ to measure the average accumulative reward staring from state $s$ by following policy $\pi$. We define $Q$-function $Q^\pi(s, a) : \mathcal{S} \times \mathcal{A} \to \mathbb{R}$ as $Q^\pi(s, a) = \mathbf{E}[\sum_{t=0}^\infty \gamma^t r_t | s_0 = s, a_0 = a]$ to represent the expected return when the action $a$ is chosen at the state $s$. By using the transition matrix and reward vector defined above, we have the compact form of the value function $\boldsymbol{v}^\pi = (\boldsymbol{I} - \gamma \boldsymbol{P}_\pi)^{-1} \boldsymbol{r}_\pi$, where $\boldsymbol{I} \in \mathbb{R}^{n \times n}$ is the identity matrix and $\boldsymbol{v}^\pi \in \mathbb{R}^n$ is the value vector with the component-wise values $[v^\pi]_s \triangleq v^\pi(s)$, with

$$v^\pi(s) \triangleq \mathbf{E}_{a \sim \pi(\cdot|s)}[Q^\pi(s, a)]. \tag{1}$$

The main objective is to find an optimal policy $\pi^*$ such that the value function is maximized, i.e.,

$$\max_\pi \mathbf{E}_{s \sim \rho_0}[v^\pi(s)] \triangleq \max_\pi \mathbf{E}_{s \sim \rho_0, a \sim \pi(\cdot|s)}[Q^\pi(s, a)]. \tag{2}$$

In what follows, we use both $Q$-function and value function $v(s)$ for convenience, and the relation between the two is given in Eqn. (2).

**Bellman Operator.** For $\boldsymbol{v} \in \mathbb{R}^n$, define the Bellman evaluation operator $T^\pi : \mathbb{R}^n \to \mathbb{R}^n$ and the Bellman operator $T : \mathbb{R}^n \to \mathbb{R}^n$ as

$$T^\pi(\boldsymbol{v}) = \boldsymbol{r}_\pi + \gamma \boldsymbol{P}_\pi \boldsymbol{v}, \ T(\boldsymbol{v}) = \max_\pi \{\boldsymbol{r}_\pi + \gamma \boldsymbol{P}_\pi \boldsymbol{v}\} = \max_\pi T^\pi(\boldsymbol{v}).$$

It is well known that the Bellman operator $T$ is a contraction mapping and has order-preserving property. Note that the Bellman operator $T$ may not be differentiable everywhere due to the $\max$ operator, and the value $\boldsymbol{v}^*$ of the optimal policy $\pi^*$ is the only fixed point of the Bellman operator $T$ (Puterman, 2014). From the definition of the Bellman Evaluation Operator $T^\pi$, we have $\boldsymbol{v}^\pi$ to be the fixed point of $T^\pi$, i.e., $\boldsymbol{v}^\pi = T^\pi(\boldsymbol{v}^\pi)$.

### 2.1 POLICY ITERATION AS NEWTON'S METHOD IN ABSTRACT DYNAMIC PROGRAMMING

Policy iteration carries out policy learning by alternating between two steps: policy improvement and policy evaluation. At time $t$, the policy evaluation step seeks to learn the value function $\boldsymbol{v}^{\pi_t}$ for the current policy $\pi_t$ by solving the fixed point equation of the Bellman evaluation operator:

$$\boldsymbol{v} = T^{\pi_t}(\boldsymbol{v}). \tag{3}$$

Denote $\boldsymbol{v}_t = \boldsymbol{v}^{\pi_t}$ for simplicity. Then in the policy improvement step, a new policy $\pi_{t+1}$ is obtained by maximizing the learnt value function $\boldsymbol{v}_t$ in the policy evaluation step, in a greedy manner, i.e.,

$$\pi_{t+1} = \arg \max_\pi T^\pi(\boldsymbol{v}_t). \tag{4}$$

Figure 1: Illustration of error propagation effect in the A-C method: The approximation errors from Critic update ($\mathcal{E}_c$) and Actor update ($\mathcal{E}_a$) are carried forward and may get amplified due to accumulation. (To distinguish the approximation errors between Critic update and Actor update, we use tilde symbol ($\widetilde{\ }$) above variables, such as policy $\widetilde{\pi}$ and value vector $\widetilde{\boldsymbol{v}}$, to represent the policy and the value vector obtained in the presence of Critic update error. We use hat symbol ($\widehat{\ }$) above the variables to represent the results with approximation error in Actor update.)

To introduce the connection between policy iteration and Newton's Method, we first define operator $F : \boldsymbol{v} \to \boldsymbol{v} - T(\boldsymbol{v})$ for convenience. As in (Grand-Clément, 2021; Puterman, 2014), $F$ can be treated as the "gradient" of an unknown function. Under the assumption that $F(\boldsymbol{v})$ is differentiable at $\boldsymbol{v}$, the Jacobian $\boldsymbol{J_v}$ of $F$ at $\boldsymbol{v}$ can be obtained as $\boldsymbol{J_v} = I - \gamma \boldsymbol{P}_{\pi(\boldsymbol{v})}$, where $\pi(\boldsymbol{v}) \triangleq \arg\max_\pi T^\pi(\boldsymbol{v})$. Note that $\boldsymbol{J_v}^{-1} = \sum_{i=1}^\infty (\gamma \boldsymbol{P}_{\pi(\boldsymbol{v})})^i$ is invertible (Puterman, 2014). Since it can be shown that $\boldsymbol{v}^{\pi_{t+1}} = (\boldsymbol{I} - \gamma \boldsymbol{P}_{\pi_{t+1}})^{-1} \boldsymbol{r}_{\pi_{t+1}} = \boldsymbol{J}_{\boldsymbol{v}^{\pi_t}}^{-1} \boldsymbol{r}_{\pi_{t+1}}$ for the policy evaluation of $\pi_{t+1}$, we have that,

$$\boldsymbol{v}^{\pi_{t+1}} = \boldsymbol{v}^{\pi_t} - \boldsymbol{J}_{\boldsymbol{v}^{\pi_t}}^{-1} \boldsymbol{J}_{\boldsymbol{v}^{\pi_t}} \boldsymbol{v}^{\pi_t} + \boldsymbol{J}_{\boldsymbol{v}^{\pi_t}}^{-1} \boldsymbol{r}_{\pi_{t+1}} = \boldsymbol{v}^{\pi_t} - \boldsymbol{J}_{\boldsymbol{v}^{\pi_t}}^{-1} F(\boldsymbol{v}^{\pi_t}), \tag{5}$$

which indicates that the analytic representation of policy iteration in the abstract dynamic programming framework reduces to Newton's Method. It is worth mentioning that the convergence behavior of policy iteration near the optimal value $\boldsymbol{v}^*$ cannot be directly obtained by using the results from convex optimization (Boyd et al., 2004) since the Bellman operator $T$ may not be differentiable at any given value vector $\boldsymbol{v}$. The full proof is included in Appendix A.

## 2.2 AN ILLUSTRATIVE EXAMPLE OF THE ERROR PROPAGATION IN ACTOR-CRITIC UPDATES

The A-C method can be viewed as a generalization of policy iteration in ADP, where the Critic update corresponds to the policy evaluation of the current policy and the Actor update performs the policy improvement. In practice, function approximation (e.g., via neural networks) is often used to approximate both the Critic and the Actor, which inevitably incurs approximation errors for the policy update and evaluation. More importantly, the approximation errors could propagate along with the iterative updates in the A-C method. We have the illustrative example to get a more concrete sense of the impact of the approximation errors on the policy update.

As illustrated in Figure 1, for a given policy $\pi_t$ with the underlying true policy value $\boldsymbol{v}^{\pi_t}$, we denote $\widetilde{\boldsymbol{v}}^{\pi_t}$ as the learnt value estimation of $\boldsymbol{v}^{\pi_t}$ in the Critic step. We further denote $\pi_{t+1}$ and $\widetilde{\pi}_{t+1}$ as the greedy policy obtained in the Actor update Eqn. (4) by using $\boldsymbol{v}^{\pi_t}$ and $\widetilde{\boldsymbol{v}}^{\pi_t}$, respectively. Let $\widehat{\pi}_{t+1}$ be the policy estimation of $\widetilde{\pi}_{t+1}$ with function approximation in the Actor step. Intuitively, $\pi_{t+1}$ is the underlying true policy update from $\pi_t$ using one step policy iteration without any error, $\widetilde{\pi}_{t+1}$ is the policy update from $\pi_t$ with approximation errors in the Critic update, and $\widehat{\pi}_{t+1}$ is the policy update from $\pi_t$ with approximation errors in both the Critic step and the Actor step. To characterize the impact of the approximation errors on the policy update, i.e., the difference between $\boldsymbol{v}^{\pi_{t+1}}$ and $\boldsymbol{v}^{\widehat{\pi}_{t+1}}$, we evaluate the Critic error, i.e., the difference between $\boldsymbol{v}^{\pi_{t+1}}$ and $\boldsymbol{v}^{\widetilde{\pi}_{t+1}}$, and the Actor error, i.e., the difference between $\boldsymbol{v}^{\widetilde{\pi}_{t+1}}$ and $\boldsymbol{v}^{\widehat{\pi}_{t+1}}$, in a separate manner. More specifically, to quantify the Critic error, we can first have the following update based on the same reasoning with Eqn. (5):

$$\boldsymbol{v}^{\widetilde{\pi}_{t+1}} = \boldsymbol{v}_t - \boldsymbol{J}_{\widetilde{\boldsymbol{v}}_t}^{-1} \left( \boldsymbol{v}_t - (\boldsymbol{r}_{\widetilde{\pi}_{t+1}} + \gamma \boldsymbol{P}_{\widetilde{\pi}_{t+1}} \boldsymbol{v}_t) \right) \triangleq \boldsymbol{v}_t - \boldsymbol{J}_{\widetilde{\boldsymbol{v}}_t}^{-1} \left( \boldsymbol{v}_t - \widetilde{T}(\boldsymbol{v}_t) \right),$$

where $\widetilde{T}(\boldsymbol{v_t}) = \boldsymbol{r}_{\widetilde{\pi}_{t+1}} + \gamma \boldsymbol{P}_{\widetilde{\pi}_{t+1}} \boldsymbol{v}_t$ and $\boldsymbol{J}_{\widetilde{\boldsymbol{v}}_t} = \boldsymbol{I} - \gamma \boldsymbol{P}_{\widetilde{\pi}_{t+1}}$. Denote the approximation error (random variable) in the Bellman operator $T$ and the Jacobian $\boldsymbol{J_v}$ by $\mathcal{E}_{T,t}$ and $\mathcal{E}_{J,t}$, i.e.,

$$\widetilde{T}(\boldsymbol{v}_t) - T(\boldsymbol{v}_t) = (\boldsymbol{r}_{\widetilde{\pi}_{t+1}} + \gamma \boldsymbol{P}_{\widetilde{\pi}_{t+1}} \boldsymbol{v}_t) - (\boldsymbol{r}_{\pi_{t+1}} + \gamma \boldsymbol{P}_{\pi_{t+1}} \boldsymbol{v}_t) \triangleq \mathcal{E}_{T,t},$$

$$\boldsymbol{J}_{\widetilde{\boldsymbol{v}}_t}^{-1} - \boldsymbol{J}_{\boldsymbol{v}_t}^{-1} = (\boldsymbol{I} - \gamma \boldsymbol{P}_{\widetilde{\pi}_{t+1}})^{-1} - (\boldsymbol{I} - \gamma \boldsymbol{P}_{\pi_{t+1}})^{-1} \triangleq \mathcal{E}_{J,t},$$

where it is clear that both error terms stem from the function approximation errors in the Critic update. To quantify the Actor error, we assume that

$$\boldsymbol{v}^{\widehat{\pi}_{t+1}} = \boldsymbol{v}^{\widetilde{\pi}_{t+1}} + \mathcal{E}_{a,t}, \tag{6}$$

where $\mathcal{E}_{a,t}$ is the error term. Therefore, by casting the A-C method as Newton's method with perturbation, we can characterize the approximation errors on the policy update:

$$\boldsymbol{v}^{\hat{\pi}_{t+1}} = \boldsymbol{v}^{\pi_{t+1}} + \mathcal{E}_{c,t} + \mathcal{E}_{a,t},$$

where $\mathcal{E}_{c,t} \triangleq -\mathcal{E}_{J,t}(\boldsymbol{v}_t - T(\boldsymbol{v}_t)) + (\boldsymbol{J}_{\boldsymbol{v}_t}^{-1} + \mathcal{E}_{J,t})\mathcal{E}_{T,t}$ and $\mathcal{E}_{a,t}$ capture the impact of the approximation error from Critic update step and Actor update step, respectively. Intuitively, as illustrated in Figure 1, both errors from the previous update in the A-C method may propagate to the next update and thus affect the convergence behavior of the algorithm substantially, in contrast to idealized policy iteration without approximation errors. This phenomenon has also been observed in the empirical results (Fujimoto et al., 2018; Thrun & Schwartz, 1993). In this work, we strive to systematically analyze the impact of the approximation errors, through (1) a detailed characterization of the approximation errors in the Critic update and the Actor update in Section 3 and (2) a thorough analysis of the error propagation effect and biases in Section 4. The illustration of our analysis is available in Appendix A.

## 3    CHARACTERIZATION OF APPROXIMATION ERRORS

**Actor-Critic Methods with Function Approximation.** In what follows, we consider that the policy is parameterized by $\theta \in \Theta$, which in general corresponds to a non-linear function class. Following (Konda & Tsitsiklis, 1999; Peters & Schaal, 2008; Kumar et al., 2019; 2020; Santos & Rust, 2004), the Q-function is parameterized by a linear function class with base function $\phi(s, a)$ and parameter $\omega \in \Omega \subset \mathbb{R}^d$, i.e., $Q_\omega(s, a) = \omega^\top \phi(s, a)$. We note that the modeling of the Q-function via linear value function is often used to extract insight in the A-C method. Similar to the policy iteration, the update in the A-C method alternates between the following two steps [1].

Critic update: The Critic updates its parameter $\omega$ to evaluate the current policy $\pi_t$, e.g., through $m$-step ($m \geq 1$) Bellman evaluation operator $T^\pi$ to the current Q-function estimator (namely, $m$-step return), which leads to the following update rule at time step $t$,

$$Q_{t+1}(s, a) \leftarrow \mathbf{E}_{\pi_t}\left[(1 - \gamma) \cdot \sum_{i=0}^{m-1} \gamma^i r(s_i, a_i) + \gamma^m \cdot Q_{\omega_t}(s_m, a_m) \mid s_0 = s, a_0 = a\right], \quad (7)$$

$$\omega_{t+1} \leftarrow \arg\min_\omega \mathbf{E}_{(s,a) \sim \rho^{\pi_t}}\left[Q_{t+1}(s, a) - \omega^\top \phi(s, a)\right]^2. \quad (8)$$

Actor update: The Actor is updated through a greedy step to maximize Q-function $Q_{\omega_{t+1}}$, i.e.,

$$\pi_{t+1} \leftarrow \arg\max_\pi \mathbf{E}_{(s,a) \sim \rho^\pi}\left[Q_{\omega_{t+1}}(s, a)\right]. \quad (9)$$

### 3.1    APPROXIMATION ERROR IN THE CRITIC UPDATE

Solving the minimization problem in Eqn. (8) involves the expectation over the stationary state-action distribution $\rho^{\pi_t}$ induced by the current policy $\pi_t$, which can be approximated by sample average in practice. Therefore, we consider the Critic update below based on two groups of samples, $\{(s_l, a_l)\}_{l=1}^N$ and $\{\tau_l\}_{l=1}^N$ where $\tau_l = \{s_{l,t}, a_{l,t}, r_{l,t}\}_{t=0}^m$, which are collected by following $\pi_t$:

$$\omega_{t+1} = \Gamma_R\Bigg\{\left(\sum_{l=1}^N \phi(s_l, a_l)\phi(s_l, a_l)^\top\right)^{-1}$$

$$\cdot \sum_{l=1}^N \left((1 - \gamma) \sum_{i=0}^{m-1} \gamma^i r_{l,i} + \gamma^m Q_{\omega_t}(s_{l,m}, a_{l,m})\right)\phi(s_{l,m}, a_{l,m})\Bigg\}, \quad (10)$$

where $\Gamma$ is the projection operator onto the Critic parameter space $\Omega$ with radius $R$ in $\mathbb{R}^d$. Since the samples in each trajectory $\tau_l$ are obtained via rolllouts, in general the samples in each trajectory follow a Markovian process (Dalal et al., 2018; Kumar et al., 2019). We further assume the samples are from the stationary distribution induced by the current policy.

In what follows, we use $\omega$ and $\widetilde{\omega}$ to distinguish the difference between the sample-based update and the solution from Eqn. (8), such that the approximation error in the Critic update can be quantified as $|Q_{\widetilde{\omega}_t} - Q_{\omega_t}|$. We first impose the following standard assumptions on the Bellman evaluation operator $T^\pi$, the base function $\phi$ and the MDP.

---

[1] We remark that our analysis framework and theoretical results are able to be applied to off-policy setting with the extra assumption on the behavior policy. We include the details in Appendix L.

**Assumption 1.** *For given Critic parameter $\omega$ and policy parameter $\theta$, the following condition holds:*
$$\inf_{\bar{\omega} \in \Omega} \mathbf{E}_{\rho^{\pi_\theta}} \left[ \left( (T^{\pi_\theta})^m Q_\omega - \bar{\omega}^\top \phi \right)(s,a) \right] = 0,$$
*where $\rho^{\pi_\theta}$ is the stationary state-action transition probability induced by policy $\pi_\theta$.*

Assumption 1 (Fu et al., 2020) indicates that the solution of the Critic update given in Eqn. (8) lies in the Critic parameter space $\Omega$. We note that this assumption is used for ease of exposition, and our results can be modified by incorporating an additional constant term when this assumption does not hold. The proof sketch in this case can be found in Appendix C.

**Assumption 2.** *The base function $\phi$ in the Critic satisfies the following two conditions: (1) $\|\phi(s,a)\|_2 \leq 1$, $\forall\ (s,a) \in \mathcal{S} \times \mathcal{A}$; and (2) the smallest singular value for $\mathbf{E}_\rho[\phi(s,a)\phi(s,a)^\top]$ is lower bounded by a positive constant $\sigma^*$ for any stationary state-action transition distribution $\rho$.*

Assumption 2 is widely used in the A-C method to guarantee that the minimization in Eqn. (8) can be attained by a unique minimizer (Fu et al., 2020; Bhandari et al., 2018; Agarwal et al., 2021).

**Assumption 3.** *The reward $r(s,a)$ satisfies the following two conditions: (1) The reward is upper bounded by a positive constant $r_{\max}$ for all $(s,a) \in \mathcal{S} \times \mathcal{A}$; and (2) the stationary state-action transition matrix $\boldsymbol{P}^\pi$ has non-zero spectral gap $1 - \lambda_\pi > 0$ for all $\pi$.*

The first condition in Assumption 3 is often used for discounted MDPs to ensure a finite value function (Thrun & Schwartz, 1993; Fujimoto et al., 2018; Fu et al., 2020). Moreover, since the samples in the same trajectories are generally correlated, the second condition is adopted to guarantee the concentration properties of the Markov chain, which is generally true for the stationary Markov chain (Jiang et al., 2018; Ortner, 2020; Amit et al., 2020).

For any $\lambda \in [0,1)$, let $\alpha_1(\lambda) = (1+\lambda)/(1-\lambda)$, $\alpha_2(\lambda) = 5/(1-\lambda)$ where $\alpha_2(0) = 1/3$. Define
$$\tilde{r}_m = \frac{\sqrt{\alpha_2^2 r_{\max}^2 \left(\max\{\lambda_{\pi_t}, 0\}\right)^2 \ln^2 p - 2m\alpha_1\left(\max\{\lambda_{\pi_t}, 0\}\right) \ln p} - \alpha_2\left(\max\{\lambda_{\pi_t}, 0\}\right) \ln p}{m} + r_{\max}.$$

Then we can have the following main result on the approximation error in the Critic update step.

**Proposition 1** (Approximation Error in Critic Update). Under Assumptions 1, 2, 3, the following inequality holds with probability at least $1 - p$, for any $t > 0$, $(s,a) \in \mathcal{S} \times \mathcal{A}$:

$$\mathbf{E}[|Q_{\omega_t}(s,a) - Q_{\tilde{\omega}_t}(s,a)|] \leq \frac{4((1-\gamma)\tilde{r}_m + \gamma^m R)}{\sqrt{N}(\sigma^*)^2} \left( -\frac{2}{3N} \log \frac{p}{4d} + \sqrt{\frac{4}{9N^2} \log^2 \frac{p}{4d} - \frac{2}{N} \log \frac{p}{4d}} \right),$$

where $d$ is the dimension of the Critic parameter $\omega$ and $R$ is the radius of Critic parameter space $\Omega$ as defined in Eqn. (10).

Proposition 1 establishes the upper bound for the approximation error in the Critic update, which encapsulates the impact of the finite sample size and the finite-step rollout with Bellman evaluation operator $T^\pi$. It can be seen from Proposition 1 that in order to obtain an accurate evaluation of the policy, we can increase the sample size $N$ in the update Eqn. (10) and have more steps of rollout with Bellman evaluation operator $T^\pi$. We remark that Proposition 1 considers the correlation across samples, and we appeal to the recent advances in Bernstein's Inequality for Markovian samples (Jiang et al., 2018)(Fan et al., 2021b) to tackle this challenge. The proof of Proposition 1 can be found in Appendix B and Appendix C.

### 3.2 APPROXIMATION ERROR IN THE ACTOR UPDATE

In practice, the greedy search step for solving Eqn. (9) is generally approximated by multiple (e.g., $N_a$) steps of policy gradient. Based on the policy gradient theorem (Silver et al., 2014; Sutton et al., 1999), we can have the following update at gradient step $k \in [1, N_a]$ in the $t$-th Actor update:
$$\theta_{t,1} = \theta_t, \ \ \theta_{t,N_a} = \theta_{t+1},$$
$$\theta_{t,k+1} = \theta_{t,k} + \alpha \mathbf{E}_{(s,a) \sim \rho^{\pi_{\theta_{t,k}}}} [Q_{\omega_{t+1}}(s,a)\nabla_\theta \pi_{\theta_{t,k}}(a|s)], \tag{11}$$
where $\alpha$ is the learning rate. For simplicity, we drop the subscript $t$ in $\theta_{t,k}$ when no confusion will arise and denote $\rho^k := \rho^{\pi_{\theta_k}}$. As in the Critic update, we sample a trajectory with length $l$ by following the current policy $\pi_{\theta_k}$, i.e., $\{s_1, a_1, s_2, a_2, \cdots, s_l, a_l\}$, to approximate the expectation in Eqn. (11). Then we can have that
$$\theta_{k+1} = \theta_k + \alpha \frac{1}{l} \sum_{i=1}^{l} [Q_{\omega_{t+1}}(s_i, a_i)\nabla_\theta \pi_{\theta_k}(a_i|s_i)] := \theta_k + \alpha(C_{k,t,1} + C_{k,t,2}) + \alpha f_{k,t}, \tag{12}$$

where $C_{k,t,1}$, $C_{k,t,2}$ and $f_{k,t}$ are defined as follows

$$C_{k,t,1} := 1/l \sum_{i=1}^{l} (Q_{\omega_{t+1}}(s_i, a_i) \nabla_\theta \pi_{\theta_k}(a_i|s_i) - Q_{\widetilde{\omega}_{t+1}}(s_i, a_i) \nabla_\theta \pi_{\theta_k}(a_i|s_i)),$$

$$C_{k,t,2} := 1/l \sum_{i=1}^{l} (Q_{\widetilde{\omega}_{t+1}}(s_i, a_i) \nabla_\theta \pi_{\theta_k}(a_i|s_i) - Q^{\pi_{\theta_t}}(s_i, a_i) \nabla_\theta \pi_{\theta_k}(a_i|s_i)),$$

$$f_{k,t} := 1/l \sum_{i=1}^{l} Q^{\pi_{\theta_t}}(s_i, a_i) \nabla_\theta \pi_{\theta_k}(a_i|s_i).$$

Here $C_{k,t,1}$ captures the error resulted from using samples to estimate expectation in the Critic update. Based on our result in Proposition 1, with high probability, this term will go to $0$ when we have infinite samples or infinite rollout length $m$. Note that $(T^{\pi_{\theta_t}})^m Q_{\omega_t} = Q_{\widetilde{\omega}_{t+1}}$ (Critic update) and $\lim_{m \to \infty} (T^{\pi_{\theta_t}})^m Q_{\omega_t} = Q^{\pi_{\theta_t}}$. And $C_{k,t,2}$ implies the approximation error when applying the Bellman operator limited ($m$) times. This term will go to $0$ when $m \to \infty$. $f_{k,t}$ is an unbiased estimation of the gradient of $\mathbf{E}_{(s,a) \sim \rho^k}[Q^{\pi_{\theta_t}}(s,a)]$, i.e., $\mathbf{E}[f_{k,t}] = \mathbf{E}_{(s,a) \sim \rho^k}[Q^{\pi_{\theta_t}}(s,a) \nabla_\theta \pi_{\theta_k}(a|s)]$.

Based on Eqn. (12), it is clear that the Actor update with the approximation error resulted from the Critic update can be viewed as a stochastic gradient update with some perturbation $C_{k,t} = C_{k,t,1} + C_{k,t,2}$. Denote $\theta_{t+1}^*$ as the solution of the Eqn. (9). For ease of exposition, we use $h(\omega, \theta)$ to denote the objective function in the Actor update:

$$h(\omega, \theta) = \mathbf{E}_{(s,a) \sim \rho^{\pi_\theta}}[Q_\omega(s,a)] = \mathbf{E}_{s \sim d^{\pi_\theta}}[v^{\pi_\omega}(s)]. \tag{13}$$

Note that $h$ is a function of Actor parameter $\theta$ for a given Critic parameter $\omega$. Next, we quantify the approximation error in the Actor update in terms of the gap $h(\omega_{t+1}, \theta_{t+1}^*) - h(\omega_{t+1}, \theta_{t+1})$.

Recall that Proposition 1 gives upper bound for the approximation error $\mathbf{E}_\omega[|Q_{\omega_t} - Q_{\widetilde{\omega}_t}|]$ in the Critic update, which has direct impact on $C_{k,t}$. Based on Proposition 1, we have the following two lemmas for the upper bounds on the bias term $b = \mathbf{E}[C_{k,t}]$ and the error term $\beta = f_{k,t} + C_{k,t} - \mathbf{E}[C_{k,t}]$ in the stochastic gradient update Eqn. (12), respectively. The proof of Lemmas 1 and 2 can be found in Appendix D and E, respectively.

**Lemma 1** ($\sigma^2$-bounded noise). *Suppose Assumptions 1, 2, 3 hold. Then with probability at least $1 - p$, $\mathbf{E}[\|\beta\|^2] \leq \|\nabla_\theta h(\omega, \theta) + b\|^2 + \sigma^2$, $\forall \theta$, where $\sigma^2 \geq 0$ is a constant and depends on $p$.*

**Lemma 2** ($\zeta^2$-bounded bias). *Suppose Assumptions 1, 2, 3 hold. Then with probability at least $1 - p$, $\|b\|^2 \leq \zeta^2$, $\forall \theta$, where $\zeta^2 \geq 0$ is a constant and depends on $p$.*

Denote the score function $\psi_\theta(a|s) := \nabla_\theta \pi_\theta(a|s)$ and we impose the following standard assumptions.

**Assumption 4.** *For any $\theta, \theta' \in \mathbb{R}^d$ and state-action pair $(s,a) \in \mathcal{S} \times \mathcal{A}$, there exist positive constants $L_\psi$, $C_\psi$ and $C_\pi$ such that the following holds: (1) $\|\psi_\theta - \psi_{\theta'}\| \leq L_\psi \|\theta - \theta'\|$; (2) $\|\psi_\theta\| \leq C_\psi$ and (3) $\|\pi_\theta(\cdot|s) - \pi_{\theta'}(\cdot|s)\|_{TV} \leq C_\pi \|\theta - \theta'\|$, where $\|\cdot\|_{TV}$ is the total-variation distance.*

We remark that the smoothness and bounded property of the score function as stated in the (1) and (2) in Assumption 4 are widely adopted in the literature (Xu et al., 2020a;b; Zou et al., 2019; Agarwal et al., 2020; Kumar et al., 2019), and it has been shown (Xu et al., 2020a) that (3) in Assumption 4 can be satisfied for any smooth policy with bounded action space.

For the sake of tractability, we next give the following two lemmas about the smoothness and Polyak-Lojasiewicz Condition on the objective function $h(\cdot, \theta)$. The proof can be found in Appendix F.

**Lemma 3** ($L$-smoothness). *Suppose Assumption 4 hold. Then function $h(\cdot, \theta)$ is bounded from below by an infimum $h^{\inf} \in \mathbb{R}$, differentiable and $\nabla h$ is $L$-Lipschitz, i.e., $\|\nabla h(\omega, \theta) - \nabla h(\omega, \theta')\| \leq L\|\theta - \theta'\|$, $\forall \omega, \theta, \theta'$.*

**Lemma 4** ($\mu$-PL). *If $\nabla h(\omega, \theta) \neq 0$, then we have $\frac{1}{2}\|\nabla h(\omega, \theta)\| \geq \mu(h(\omega, \theta^*) - h(\omega, \theta)) \geq 0, \forall \theta, \omega$.*

Let $\alpha \leq \frac{1}{2L}$. Next we present the upper bound of the approximation error in the Actor update.

**Proposition 2** (Approximation Error in Actor Update). Given the updated Actor parameter $\theta_{t-1}$, the following inequality holds with probability at least $1 - p$:

$$\mathbf{E}_\theta[\|h(\omega, \theta_t^*) - h(\omega, \theta_t)\|] \leq (1 - \alpha\mu)^{N_a}(h(\omega, \theta_t^*) - h(\omega, \theta_{t-1})) + \frac{\zeta^2 + 2\alpha L\sigma^2}{2\mu},$$

where $\sigma^2$, $\zeta^2$, $L$ and $\mu$ are defined in Lemma 1, Lemma 2, Lemma 3 and Lemma 4, respectively.

Proposition 2 reveals that due to the bias and noise induced by the Critic approximation error, running more gradient iterations do not necessarily guarantee the convergence to the optimal policy $\pi_{\theta_t^*}$. Note that Lemmas 1 and 2 in (Ajalloeian & Stich, 2020) are given in the form of assumptions, whereas in this work, we justify that both assumptions hold with high probability and prove Proposition 2, and the proof can be found in Appendix G.

# 4 IMPACT OF APPROXIMATION ERRORS AND WARM-START POLICY ON FINITE-TIME LEARNING PERFORMANCE

We next quantify the impact of the approximations errors on the sub-optimality gap of the Warm-Start A-C method with inaccurate Actor/Critic updates. We first cast the A-C method as Newton's Method with perturbation, and then present both the finite-time upper bound and lower bound on the finite-time learning performance in Section 4.1.

**Actor-Critic Method as Newton's Method with Perturbation.** As mentioned earlier, the Critic update follows Eqn. (10) with finite samples and finite step rollout with Bellman evaluation operator $T^\pi$ and the Actor update follows Eqn. (12). Given the policy $\pi_t$ at time $t$, we denote the resulting policy of one A-C update as $\hat\pi_{t+1}$. Recall that we use $\widetilde\pi_{t+1}$ to denote the policy attained the $\max$ in $T(\boldsymbol{v}^{\pi_t})$ as illustrated in Figure 1. Furthermore, we define the following notations for ease of our discussion: (1) Denote $\mathcal{E}_{v,t} = \boldsymbol{v}^{\hat\pi_{t+1}} - \boldsymbol{v}^{\widetilde\pi_{t+1}}$ as the approximation error in the Actor update; (2) Denote $\mathcal{E}_{r,t} = \boldsymbol{r}_{\widetilde\pi_{t+1}} - \boldsymbol{r}_{\hat\pi_{t+1}}$ as the error in the reward vector, which is induced by the approximation error in the Actor update step; (3) Denote $\mathcal{E}_{P,t} = \boldsymbol{P}_{\widetilde\pi_{t+1}} - \boldsymbol{P}_{\hat\pi_{t+1}}$ as the error in the transition matrix $\boldsymbol{P}$; (4) Denote $\mathcal{E}_{\hat{J},t} = \boldsymbol{J}_{\widetilde{\boldsymbol{v}}_t}^{-1} - \boldsymbol{J}_{\hat{\boldsymbol{v}}_t}^{-1}$ where $\boldsymbol{J}_{\hat{\boldsymbol{v}}_t} = \boldsymbol{I} - \gamma\boldsymbol{P}_{\hat\pi_{t+1}}$ and $\boldsymbol{J}_{\widetilde{\boldsymbol{v}}_t} = \boldsymbol{I} - \gamma\boldsymbol{P}_{\widetilde\pi_{t+1}}$.

Following the same line as in Section 2.2, we treat the A-C algorithm as Newton's method with perturbation $\mathcal{E}_t$, i.e.,

$$\boldsymbol{v}^{\hat\pi_{t+1}} = \boldsymbol{v}^{\hat\pi_t} - (\boldsymbol{J}_{\hat{\boldsymbol{v}}_t}^{-1}(\boldsymbol{v}^{\hat\pi_t} - T(\boldsymbol{v}^{\hat\pi_t})) - \mathcal{E}_t) := \boldsymbol{v}^{\hat\pi_t} - \hat{\mathcal{L}}(t), \tag{14}$$

where $\hat{\mathcal{L}}(t)$ is the stochastic estimator of Newton's update $\mathcal{L}(t) = \boldsymbol{J}_{\hat{\boldsymbol{v}}^{\pi_t}}^{-1}\left(\boldsymbol{v}^{\hat\pi_t} - T\left(\boldsymbol{v}^{\hat\pi_t}\right)\right)$, and

$$\mathcal{E}_t = \mathcal{E}_{v,t} + \mathcal{E}_{\hat{J},t}(\boldsymbol{v}^{\hat\pi_{t+1}} - (\boldsymbol{r}_{\widetilde\pi_{t+1}} + \gamma\boldsymbol{P}_{\widetilde\pi_{t+1}}\boldsymbol{v}^{\hat\pi_{t+1}})) - \boldsymbol{J}_{\hat{\boldsymbol{v}}_t}^{-1}(\mathcal{E}_{r,t} + \gamma\mathcal{E}_{P,t}\boldsymbol{v}^{\hat\pi_t}),$$

which can be further decomposed into bias and Martingale difference noise as follows:

$$\mathcal{B}(t) \triangleq \mathbf{E}[\hat{\mathcal{L}}(t)] - \mathcal{L}(t) = \mathbf{E}[\mathcal{E}_t],$$

$$\mathcal{N}(t) \triangleq \hat{\mathcal{L}}(t) - \mathbf{E}[\hat{\mathcal{L}}(t)] = \mathcal{E}_t - \mathbf{E}[\mathcal{E}_t].$$

We have a few observations in order. It can be seen that the perturbation $\mathcal{E}_t$ results from both Actor approximation error (e.g., $\mathcal{E}_{r,t}$, $\mathcal{E}_{P,t}$) and Critic approximation error (e.g., $\mathcal{E}_{v,t}$). More importantly, the learnt $Q$ function in the Critic update Eqn. (10) is biased in general due to finite rollout steps $m$, which further leads to the biased gradients in the Actor update Eqn. (11) (Kumar et al., 2019). Clearly, the estimation bias plays an important role in affecting the learning performance, especially when deep neural networks are used as function approximations, which has been extensively investigated using empirical studies (Fujimoto et al., 2018; Elfwing et al., 2018; Van Hasselt et al., 2016).

Next, we examine the bias $\mathcal{B}(t)$ based on the approximation errors in the Actor/Critic updates. Combining the results in Proposition 1 and 2 on the approximation error in the Critic/Actor updates, we have the following result on the bias $\mathcal{B}(t)$. A full derivation is given in Appendix H.

**Proposition 3** (Upper Bound on the Bias)**.** Suppose Assumption 4 holds. Let $S_\epsilon(\cdot)$ be an open ball of radius $\epsilon$. There exist positive constants $L_b$, and $\epsilon$, such that when $\theta_{t+1} \in S_\epsilon(\theta_{t+1}^*)$, the following holds for any $t > 0$,

$$\|\mathcal{B}(t)\| \le L_b\mathbf{E}_\theta[\|h(\cdot, \theta_{t+1}^*) - h(\cdot, \theta_{t+1})\|].$$

## 4.1 FINITE-TIME LEARNING PERFORMANCE AND ERROR PROPAGATION EFFECT

**Lower Bound on Performance Gap.** **Aiming to understand "whether online learning can be accelerated by a warm-start policy",** we first derive a lower bound to quantify the impact of the bias and the error propagation. By unrolling the recursion of the Newton update (with perturbation) Eqn. (14), we obtain the following theorem.

**Theorem 1.** *The lower bound of $\|\mathbf{E}[\boldsymbol{v}^* - \boldsymbol{v}^{\hat\pi_{t+1}}]\|$ satisfies that*

$$\|\mathbf{E}\left[\boldsymbol{v}^* - \boldsymbol{v}^{\hat\pi_{t+1}}\right]\| \ge \|\gamma^{t+1}\bar{\boldsymbol{P}}_{t+1}(\boldsymbol{v}^* - \boldsymbol{v}^{\pi_0}) + \sum_{i=1}^t \gamma^i\bar{\boldsymbol{P}}_i\mathcal{B}(t-i) + \mathcal{B}(t)\|, \tag{15}$$

*where $\bar{\boldsymbol{P}}_{t+1} = \mathbf{E}\left[\left(\prod_{i=0}^t \boldsymbol{P}_{\pi_{t+1-i}}\right)\right]$ and $\pi_0$ is the warm-start policy.*

**Error Propagation and Accumulation.** It can be seen form Theorem 1 that the bias terms $\{\mathcal{B}(t)\}$ add up over time, and the propagation effect of the bias terms is encapsulated by the last two terms on the right side of Eqn. (15). Clearly, the first term on the right side, corresponding to the impact of

the warm-start policy $\pi_0$, diminishes with A-C updates. To get a more concrete sense of Theorem 1, we consider the following special settings. (1) When the bias is always positive, i.e., $\mathcal{B}(t) > \mathbf{0}$ for all $t \geq 0$, the lower bound in Theorem 1 is always positive, i.e., $\|\mathbf{E}\left[\boldsymbol{v}^* - \boldsymbol{v}^{\hat{\pi}_{t+1}}\right]\| \geq \|\mathcal{B}(t)\| > 0$. In this case, the sub-optimal gap remains bounded away from zero. Similar conclusion can be made when the bias is always negative. (2) When the bias term can be either positive or negative, the lower bound is shown as Eqn. (15). In this case, the learning performance of the A-C algorithm largely depends on the behavior of the Bias term. It can be seen from Theorem 1 that even when the warm-start policy is near-optimal, it is still challenging to guarantee that online fine-tuning can improve the policy if the approximation error is not handled correctly. We note that this has also been observed empirically (Nair et al., 2020; Lee et al., 2022).

**Upper Bound on Performance Gap.**   In order to derive the upper bound on the sub-optimality gap, we first introduce the following standard assumption on the Jacobian $\boldsymbol{J_v}$ as in the analysis of policy iteration (Puterman & Brumelle, 1979; Grand-Clément, 2021).

**Assumption 5.** *For some $q > 0$ there exists a constant $0 < L_J < +\infty$ such that*
$$\|\boldsymbol{J}_{\boldsymbol{v}^\pi} - \boldsymbol{J}_{\boldsymbol{v}^*}\| \leq L_J \|\boldsymbol{v}^\pi - \boldsymbol{v}^*\|^q \ \ \forall \, \pi,$$
*and there is a constant $0 < M < +\infty$ such that $\|\boldsymbol{J}_{\boldsymbol{v}^\pi}^{-1}\| \leq M, \ \ \forall \, \pi$.*

Denote $H_t := \|\boldsymbol{J}_{\hat{\boldsymbol{v}}_t}^{-1}\left[\boldsymbol{J}_{\hat{\boldsymbol{v}}_t} - \boldsymbol{J}_{\boldsymbol{v}^*}\right]\|$. Clearly, we have $H_t$ is upper bounded by $H_t \leq ML_J \|\boldsymbol{v}^{\hat{\pi}_t} - \boldsymbol{v}^*\|^q$ from Assumption 5. Next, We present the finite-time upper bound in Theorem 2.

**Theorem 2.** *Suppose Assumption 5 holds. Then we have that for any $t > 0$,*
$$\|\mathbf{E}[\boldsymbol{v}^{\hat{\pi}_{t+1}} - \boldsymbol{v}^*]\| \leq \left(\prod_{i=0}^{t} H_{t-i}\right) \|\boldsymbol{v}^* - \boldsymbol{v}^{\pi_0}\| + \sum_{i=1}^{t}\left(\prod_{j=1}^{i} H_{t-j}\right) \|\mathcal{B}(t-i)\| + \|\mathcal{B}(t)\|.$$

**Under what conditions online learning can be accelerated by the warm-start policy?** The upper bound in Theorem 2 sheds light on the impact of warm-start policy $\pi_0$ (the first term) and the bias $\{\mathcal{B}(t)\}$ (the last two terms), thereby providing guidance on how to achieve desired finite-time learning performance. Specifically, consider the case when the approximation error is unbiased. Clearly, we have $\|\mathbf{E}[\boldsymbol{v}^{\hat{\pi}_{t+1}} - \boldsymbol{v}^*]\| \leq \left(\prod_{i=0}^{t} H_{t-i}\right) \|\boldsymbol{v}^* - \boldsymbol{v}^{\pi_0}\|$, which decreases quickly if the warm-start policy $\pi_0$ is close to the optimal policy $\pi^*$. This observation corroborates the most recent empirically finding, where the online RL is able to further improve the warm-start policy by few adaptation steps (Silver et al., 2017; Bertsekas, 2022a; Kalashnikov et al., 2018). More generally, when the bias $\mathcal{B}(t) \neq \mathbf{0}$, the upper bound hinges heavily on the biases in the approximation errors, even when the warm-start policy $\pi_0$ is close to the optimal policy. In this case, recall the result on the upper bound of the bias $\mathcal{B}(t)$ in proposition 3, where we establish the connection between the bias and the approximation error. As expected, in order to reduce the performance gap, it is essential to decrease the bias in the approximation error, which can be achieved by increasing gradient steps and sample sizes. The proof of Theorem 1 and 2 are relegated to the Appendix I and J, respectively. The experiments results and analysis on the Gridworld benchmark can be found in Appendix K.

## 5   CONCLUSION

In this work, we take a finite-time analysis approach to quantify the impact of approximation errors on the learning performance of Warm-Start A-C method with a given prior policy. By delving into the intricate coupling between the updates of the Actor and the Critic, we first provide upper bounds on the approximation errors in both the Critic update and Actor update of online adaptation, respectively, where the recent advances on Bernstein's Inequality are leveraged to deal with the sample correlation therein. Based on these results, we next cast the Warm-Start A-C method as Newton's method with perturbation, which serves as the foundation for characterizing the impact of the approximation errors on the finite-time learning performance of Warm-Start A-C. In particular, we derive lower bounds on the sub-optimality gap under biased approximation errors, indicating that the performance gap can be bounded away from zero even for Warm-Start A-C with a good prior policy. And we also provide upper bounds on the sub-optimality gap, which provides guidance on the design of Warm-Start RL for achieving desired finite-time learning performance.

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

# Appendix

## A   EXAMPLES IN SECTION 2.2

In this section, we elaborate further on the illustrative example in Section 2.2. We use the notation defined in Figure 1.

**Policy Iteration as Newton's Method.** Based on (Puterman & Brumelle, 1979)(Grand-Clément, 2021), we first build the relation between policy iteration and Newton's Method in the abstract dynamic programming (ADP) framework, assuming accurate updates.

From the definition of the value function $\boldsymbol{v}$, we have that for any policy $\pi$,
$$\boldsymbol{v}^{\pi} = \boldsymbol{r}_{\pi} + \gamma \boldsymbol{P}_{\pi} \boldsymbol{v}^{\pi}.$$
Recall the definition of Bellman evaluation operator $T^{\pi}(\cdot)$ and the Bellman operator $T(\cdot)$,
$$T^{\pi}(\boldsymbol{v}) = \boldsymbol{r}_{\pi} + \gamma \boldsymbol{P}_{\pi} \boldsymbol{v}, \ T(\boldsymbol{v}) = \max_{\pi}\{\boldsymbol{r}_{\pi} + \gamma \boldsymbol{P}_{\pi} \boldsymbol{v}\} = \max_{\pi} T^{\pi}(\boldsymbol{v}).$$
It follows that
$$\begin{aligned}
\boldsymbol{v}^{\pi_{t+1}} &= \boldsymbol{J}_{\boldsymbol{v}^{\pi_t}}^{-1} \boldsymbol{r}_{\pi_{t+1}} \\
&= \boldsymbol{v}^{\pi_t} - \boldsymbol{v}^{\pi_t} + \boldsymbol{J}_{\boldsymbol{v}^{\pi_t}}^{-1} \boldsymbol{r}_{\pi_{t+1}} \\
&= \boldsymbol{v}^{\pi_t} - \boldsymbol{J}_{\boldsymbol{v}^{\pi_t}}^{-1} \boldsymbol{J}_{\boldsymbol{v}^{\pi_t}} \boldsymbol{v}^{\pi_t} + \boldsymbol{J}_{\boldsymbol{v}^{\pi_t}}^{-1} \boldsymbol{r}_{\pi_{t+1}} \\
&= \boldsymbol{v}^{\pi_t} - \boldsymbol{J}_{\boldsymbol{v}^{\pi_t}}^{-1} \left( -\boldsymbol{r}_{\pi_{t+1}} + \boldsymbol{J}_{\boldsymbol{v}^{\pi_t}} \boldsymbol{v}^{\pi_t} \right) \\
&= \boldsymbol{v}^{\pi_t} - \boldsymbol{J}_{\boldsymbol{v}^{\pi_t}}^{-1} \left( -\boldsymbol{r}_{\pi_{t+1}} + \left( \boldsymbol{I} - \gamma \boldsymbol{P}_{\pi_{t+1}} \right) \boldsymbol{v}^{\pi_t} \right) \\
&= \boldsymbol{v}^{\pi_t} - \boldsymbol{J}_{\boldsymbol{v}^{\pi_t}}^{-1} \left( \boldsymbol{v}^{\pi_t} - \boldsymbol{r}_{\pi_{t+1}} - \gamma \boldsymbol{P}_{\pi_{t+1}} \boldsymbol{v}^{\pi_t} \right) \\
&= \boldsymbol{v}^{\pi_t} - \boldsymbol{J}_{\boldsymbol{v}^{\pi_t}}^{-1} \left( \boldsymbol{v}^{\pi_t} - T\left( \boldsymbol{v}^{\pi_t} \right) \right), \quad\quad\quad (16)
\end{aligned}$$
where $\boldsymbol{J}_{\boldsymbol{v}} = \boldsymbol{I} - \gamma \boldsymbol{P}_{\pi(\boldsymbol{v})}$ and $\pi(\boldsymbol{v})$ attains the $\max$ in $T(\boldsymbol{v})$. Eqn. (16) establishes a connection between policy iteration under ADP and Newton's Method. Specifically, if we assume function $F : \boldsymbol{v} \to \boldsymbol{v} - T(\boldsymbol{v})$ is differentiable at any vector $\boldsymbol{v}$ visited by policy iteration, then we have $\boldsymbol{v}_{t+1} = \boldsymbol{v}_t + \boldsymbol{J}_{\boldsymbol{v}_t}^{-1} F(\boldsymbol{v}_t)$, which is exactly the update of the Newton's Method in convex optimization (Boyd et al., 2004). Due to the fact that $F(\cdot)$ may not be differentiable at all $\boldsymbol{v}$ in policy iteration, the assumptions on the Lipschitzness of $\boldsymbol{v} \to \boldsymbol{J}_{\boldsymbol{v}}$ is commonly used to prove the convergence of the policy iteration (see Assumption 5). Following the same line, next we show the case when function approximation is used in the A-C algorithm.

**A-C Updates with Function Approximation.** Consider the illustration example in Section 2.2. Next we outline the main differences between the A-C update with function approximation and the policy iteration in the ADP framework, and cast A-C based policy iteration with function approximation as Newton's Method with perturbation. Specifically,
$$\begin{aligned}
\boldsymbol{v}^{\widetilde{\pi}_{t+1}} &= \boldsymbol{J}_{\boldsymbol{v}^{\widetilde{\pi}_t}}^{-1} \boldsymbol{r}_{\widetilde{\pi}_{t+1}} \\
&= \boldsymbol{v}^{\pi_t} - \boldsymbol{v}^{\pi_t} + \boldsymbol{J}_{\boldsymbol{v}^{\widetilde{\pi}_t}}^{-1} \boldsymbol{r}_{\widetilde{\pi}_{t+1}} \\
&= \boldsymbol{v}^{\pi_t} - \boldsymbol{J}_{\boldsymbol{v}^{\widetilde{\pi}_t}}^{-1} \boldsymbol{J}_{\boldsymbol{v}^{\widetilde{\pi}_t}} \boldsymbol{v}^{\pi_t} + \boldsymbol{J}_{\boldsymbol{v}^{\widetilde{\pi}_t}}^{-1} \boldsymbol{r}_{\widetilde{\pi}_{t+1}} \\
&= \boldsymbol{v}^{\pi_t} - \boldsymbol{J}_{\boldsymbol{v}^{\widetilde{\pi}_t}}^{-1} \left( -\boldsymbol{r}_{\widetilde{\pi}_{t+1}} + \boldsymbol{J}_{\boldsymbol{v}^{\widetilde{\pi}_t}} \boldsymbol{v}^{\pi_t} \right) \\
&= \boldsymbol{v}^{\pi_t} - \boldsymbol{J}_{\boldsymbol{v}^{\widetilde{\pi}_t}}^{-1} \left( -\boldsymbol{r}_{\widetilde{\pi}_{t+1}} + \left( \boldsymbol{I} - \gamma \boldsymbol{P}_{\widetilde{\pi}_{t+1}} \right) \boldsymbol{v}^{\pi_t} \right) \\
&= \boldsymbol{v}^{\pi_t} - \boldsymbol{J}_{\boldsymbol{v}^{\widetilde{\pi}_t}}^{-1} \left( \boldsymbol{v}^{\pi_t} - \left( \boldsymbol{r}_{\widetilde{\pi}_{t+1}} + \gamma \boldsymbol{P}_{\widetilde{\pi}_{t+1}} \boldsymbol{v}^{\pi_t} \right) \right) \\
&\triangleq \boldsymbol{v}^{\pi_t} - \boldsymbol{J}_{\boldsymbol{v}^{\widetilde{\pi}_t}}^{-1} \left( \boldsymbol{v}^{\pi_t} - \widetilde{T}\left( \boldsymbol{v}^{\pi_t} \right) \right),
\end{aligned}$$
where $\boldsymbol{J}_{\boldsymbol{v}^{\widetilde{\pi}_t}} = \boldsymbol{I} - \gamma \boldsymbol{P}_{\pi(\boldsymbol{v}^{\widetilde{\pi}_t})}$ and $\pi(\boldsymbol{v})$ attains the $\max$ in $T(\boldsymbol{v})$ (not $\widetilde{T}(\boldsymbol{v})$), with the following two operators defined as
$$\begin{aligned}
T(\boldsymbol{v_t}) &\triangleq \boldsymbol{r}_{\pi_{t+1}} + \gamma \boldsymbol{P}_{\pi_{t+1}} \boldsymbol{v_t}, \\
\widetilde{T}(\boldsymbol{v_t}) &\triangleq \boldsymbol{r}_{\widetilde{\pi}_{t+1}} + \gamma \boldsymbol{P}_{\widetilde{\pi}_{t+1}} \boldsymbol{v_t}.
\end{aligned}$$

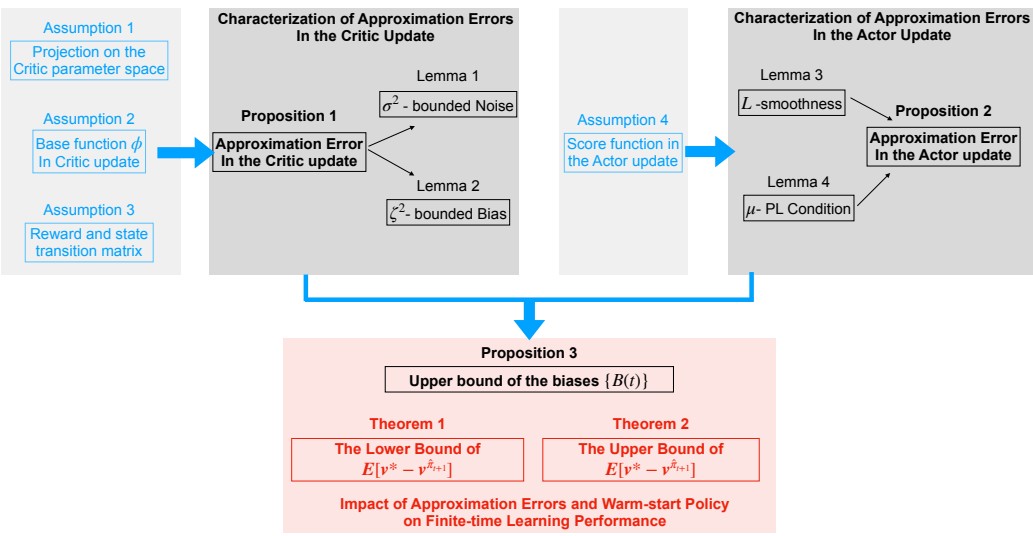

Figure 2: Illustration of the theoretical analysis.

For convenience, let $\mathcal{E}_{T,t}$ and $\mathcal{E}_{J,t}$ denote the approximation errors in the Bellman operator $T$ and the Jacobian $\boldsymbol{J_v}$, i.e.,

$$\widetilde{T}(\boldsymbol{v}_t) - T(\boldsymbol{v}_t) = (\boldsymbol{r}_{\widetilde{\pi}_{t+1}} + \gamma \boldsymbol{P}_{\widetilde{\pi}_{t+1}} \boldsymbol{v}_t) - (\boldsymbol{r}_{\pi_{t+1}} + \gamma \boldsymbol{P}_{\pi_{t+1}} \boldsymbol{v}_t) \triangleq \mathcal{E}_{T,t},$$

$$\boldsymbol{J}_{\widetilde{\boldsymbol{v}}_t}^{-1} - \boldsymbol{J}_{\boldsymbol{v}_t}^{-1} = (\boldsymbol{I} - \gamma \boldsymbol{P}_{\widetilde{\pi}_{t+1}})^{-1} - (\boldsymbol{I} - \gamma \boldsymbol{P}_{\pi_{t+1}})^{-1} \triangleq \mathcal{E}_{J,t},$$

and define

$$\boldsymbol{v}^{\hat{\pi}_{t+1}} \triangleq \boldsymbol{v}^{\widetilde{\pi}_{t+1}} + \mathcal{E}_{a,t},$$

where $\mathcal{E}_{a,t}$ capture the error induced by inaccurate policy improvement (the greedy step, e.g., Eqn. (9)) in the Actor update. Then we have that

$$\boldsymbol{v}^{\widetilde{\pi}_{t+1}} = \boldsymbol{v}^{\pi_t} - \boldsymbol{J}_{\boldsymbol{v}^{\widetilde{\pi}_t}}^{-1} \left( \boldsymbol{v}^{\pi_t} - \widetilde{T}(\boldsymbol{v}^{\pi_t}) \right)$$

$$= \boldsymbol{v}_t - (\boldsymbol{J}_{\boldsymbol{v}_t}^{-1} + \mathcal{E}_{J,t})(\boldsymbol{v}_t - T(\boldsymbol{v}_t) - \mathcal{E}_{T,t})$$

$$= \underbrace{\boldsymbol{v}_t - \boldsymbol{J}_{\boldsymbol{v}_t}^{-1}(\boldsymbol{v}_t - T(\boldsymbol{v}_t))}_{\text{Exact Newton Step}} \underbrace{-\mathcal{E}_{J,t}(\boldsymbol{v}_t - T(\boldsymbol{v}_t)) + (\boldsymbol{J}_{\boldsymbol{v}_t}^{-1} + \mathcal{E}_{J,t})\mathcal{E}_{T,t}}_{\text{Perturbation}}$$

$$\triangleq \underbrace{\boldsymbol{v}_t - \boldsymbol{J}_{\boldsymbol{v}_t}^{-1}(\boldsymbol{v}_t - T(\boldsymbol{v}_t))}_{\text{Exact Newton Step}} + \mathcal{E}_t$$

$$= \boldsymbol{v}^{\pi_{t+1}} + \mathcal{E}_t.$$

In a nutshell, we have that

$$\boldsymbol{v}^{\hat{\pi}_{t+1}} = \boldsymbol{v}^{\pi_{t+1}} + \mathcal{E}_{c,t} + \mathcal{E}_{a,t},$$

where

$$\mathcal{E}_{c,t} \triangleq -\mathcal{E}_{J,t}(\boldsymbol{v}_t - T(\boldsymbol{v}_t)) + (\boldsymbol{J}_{\boldsymbol{v}_t}^{-1} + \mathcal{E}_{J,t})\mathcal{E}_{T,t}.$$

## B    PROOF OF BERNSTEIN'S INEQUALITY WITH GENERAL MAKOVIAN SAMPLES

In this section, we provide the proof of Bernstein's Inequality with General Makovian samples following the proof in Theorem 2 (Jiang et al., 2018).

With a bit abuse of notation, let $\pi$ denote the stationary distribution of the Markov chain $\{X_i\}_{i \geq 1}$. We define $\pi(h) := \int h(x)\pi(dx)$ to be the integral of function $h$ with respect to $\pi$. Let $\mathcal{L}_2(\pi) = \{h : \pi(h^2) < \infty\}$ be the Hilbert space of square-integrable functions and $\mathcal{L}_2^0(\pi) = \{h \in \mathcal{L}_2(\pi) : \pi(h) = 0\}$ be the subspace of mean zero functions. Let $P$ be the Markov transition matrix of its

underlying (state space) graph and $P^*$ be its adjoint in the Hilbert space. Let $\lambda(P) \in [0,1]$ be the operator norm of $P$ on $\mathcal{L}_2^0(\pi)$ and $\lambda_r(P) \in [-1,1]$ be the rightmost spectral value of $(P + P^*)/2$. Then the right spectral gap of $P$ is defined as $1 - \lambda_r$ (Levin & Peres, 2017) (We remark that in Assumption 3, we assume the absolute spectral gap is non-zero, which implies the right spectral gap is also non-zero. This is true since $-1 \leq \lambda_r \leq \lambda \leq 1$.). Let $E^h$ denote the multiplication operator of function $e^h : x \mapsto e^{h(x)}$. In the Hilbert space $\mathcal{L}_2(\pi)$, we define the norm of a function $h$ to be $\|h\|_\pi = \sqrt{\langle h, h \rangle_\pi}$. Furthermore, we introduce the norm of a linear operator $T$ on $\mathcal{L}_2(\pi)$ as $\|\|T\|\|_\pi = \sup\{\|Th\|_\pi : \|h\|_\pi = 1\}$.

We first restate Bernstein's Inequality with General Makovian Samples (Jiang et al., 2018) in the following theorem. Let $\alpha_1(\lambda) = (1 + \lambda)/(1 - \lambda)$, $\alpha_2(\lambda) = 5/(1 - \lambda)$ and $\alpha_2(0) = 1/3$.

**Theorem 3** (Bernstein's Inequality with General Makovian Samples). *Suppose $\{X_i\}_{i \geq 1}$ is a stationary Markov chain with invariant distribution $\pi$ and non-zero right spectral gap $1 - \lambda_r > 0$, and $f :\mapsto x[-c, c]$ is a function with $\pi(f) = 0$. Let $\sigma^2 = \pi(f^2)$. Then, for any $0 \leq t < (1 - \max\{\lambda_r, 0\})/5c$ and any $\epsilon > 0$,*

$$\mathbf{P}_\pi \left( \frac{1}{n} \sum_{i=1}^n f(X_i) > \epsilon \right) \leq \exp \left( -\frac{n\epsilon^2/2}{\alpha_1(\max\{\lambda_r, 0\}) \cdot \sigma^2 + \alpha_2(\max\{\lambda_r, 0\}) \cdot c\epsilon} \right). \quad (17)$$

*Proof.* **Step 1.** Establish the upper bound of $\mathbf{E}\left[ e^{t \sum_i^n f_i(X_i)} \right]$.

Let $I : x \mapsto 1$ be the fucntion mapping $x$ to 1 and let $\Pi$ be the projection operator onto 1, i.e., $\Pi : g \mapsto \langle h, I \rangle_\pi I = \pi(h)I$. Define the León-Perron operator to be $\widehat{P}_\gamma = \gamma I + (1 - \gamma)\Pi$, $\gamma \in [0, 1)$. Then we recall the following lemma (Lemma 2, (Jiang et al., 2018)) on the stationary Markov chain (Fan et al., 2021a).

**Lemma 5.** *Let $\{X_i\}$ be a stationary Markov chain with invariant measure $\pi$ and non-zero right spectral gap $1 - \lambda_r > 0$. For any bounded function $f$ and any $t \in \mathbb{R}$,*

$$\mathbf{E}_\pi \left[ e^{t \sum_{i=1}^n f(X_i)} \right] \leq \left\|\left\| E^{tf/2} \widehat{P}_{\max\{\lambda_r, 0\}} E^{tf/2} \right\|\right\|_\pi^n.$$

Lemma 6 indicates that it is sufficient to prove the upper bound of $\mathbf{E}\left[ e^{t \sum_i^n f_i(X_i)} \right]$ by proving the upper bound of $\left\|\left\| E^{tf/2} \widehat{P}_{\max\{\lambda_r, 0\}} E^{tf/2} \right\|\right\|_\pi^n$.

To this end, we first invoke the following lemma (Lemma 6, (Jiang et al., 2018)) to construct $\widehat{f}_k \approx f$ such that for any $\lambda \in [0, 1)$, $\left\|\left\| E^{tf/2} \widehat{P}_\lambda E^{tf/2} \right\|\right\|_\pi = \lim_{k \to \infty} \left\|\left\| E^{t\widehat{f}_k/2} \widehat{P}_\lambda E^{t\widehat{f}_k/2} \right\|\right\|_\pi$.

**Lemma 6.** *For function $f : x \in \mathcal{X} \mapsto [-c, c]$ such that $\pi(f) = c$, $\pi(f^2) = \sigma^2$. Let $\lceil \cdot \rceil$ be the ceiling function and $\widetilde{f}_k(x) = \left\lceil \frac{f(x)+c}{c/3k} \right\rceil \times \frac{c}{3k} - c$. Let $\widehat{f}_k = \frac{\widetilde{f}_k - \pi(\widetilde{f}_k)}{1 + 1/3k}$. Then $\widetilde{f}_k$ takes at most $6k + 1$ possible values and satisfies that for any bounded linear operator $T$ acting on the Hilbert Space $\mathcal{L}_2(\pi)$ and any $t \in \mathbb{R}$,*

$$\left\|\left\| E^{tf/2} T E^{tf/2} \right\|\right\|_\pi = \lim_{k \to \infty} \left\|\left\| E^{t\widehat{f}_k/2} T E^{t\widehat{f}_k/2} \right\|\right\|_\pi.$$

Assume that the Markov chain $\{\widehat{X}_i\}_{i \geq 1}$, $\widehat{X}_i \in \mathcal{X}$ is generated by the León-Perron operator $\widehat{P}_\lambda$. It follows that $\{\widehat{Y}_i\}_{i \geq 1} = \{\widehat{f}_k(\widehat{X}_i)\}_{i \geq 1}$ is a Markov chain in the state space $\mathcal{Y} = \widehat{f}_k(\mathcal{X})$. We recall the following lemma (Lemma 7, (Jiang et al., 2018)) on the relation between the two Markov chains.

**Lemma 7.** *Let $\widehat{P}_\lambda$ be the León-Perron operator with $\lambda \in [0, 1)$ on state space $\mathcal{X}$. Let $f$ be a function on $\mathcal{X}$. On the finite state space $\mathcal{Y} = \{y \in f(\mathcal{X}) : \pi(\{x : f(x) = y\}) > 0\}$, define a transition matrix $\widehat{Q}_\lambda = \lambda I + (1 - \lambda) I \mu^\top$, with transition vector $\mu$ consisting of elements $\pi(\{x : f(x) = y\})$ for $y in \mathcal{Y}$. Let $E^{t\mathcal{Y}}$ denote the diagonal matrix with elements $e^{ty} : y \in \mathcal{Y}$. Then we have,*

$$\left\|\left\| E^{tf/2} \widehat{P}_\lambda E^{tf/2} \right\|\right\|_\pi = \left\|\left\| E^{t\mathcal{Y}/2} \widehat{Q}_\lambda E^{t\mathcal{Y}/2} \right\|\right\|_\mu.$$

Next, we bound the term $\left\|\left\|\left|E^{t\mathcal{Y}/2}\widehat{Q}_\lambda E^{t\mathcal{Y}/2}\right\|\right\|\right\|_\mu$ by the expansion of the largest eigenvalue of the perturbed Markov operator $E^{tf/2}PE^{tf/2}$ as a series in $t$. Specifically, we recall the following result (Lezaud, 1998).

**Lemma 8.** *Consider a reversible, irreducible Markov chain on finite state space $\mathcal{X}$. Let $D$ be the diagonal matrix with $\{f(x) : x \in \mathcal{X}\}$ and $T^{(m)} = PD^m/m!$ for any $m \geq 0$ with $D^0 = I$. Assume the invariant distribution of the Markov chain is $\pi$ and the second largest eigenvalue of the transition matrix $P$ is $\lambda_r < 1$. Let $t_0 = \left(2\left\|\left\|\left|T^{(1)}\right\|\right\|\right\|_\pi (1-\lambda_r)^{-1} + c_0\right)^{-1}$ for some $c_0$ such that*

$$\left\|\left\|\left|T^{(m)}\right\|\right\|\right\|_\pi \leq \left\|\left\|\left|T^{(1)}\right\|\right\|\right\|_\pi c_0^{m-1}, \forall m \geq 1.$$

*Denote the largest eigenvalue of $PE^{tf}$ by $\beta(t)$ and $Z = (I - P + \Pi)^{-1} - \Pi$. Let $Z^0 = -\Pi$, $Z^{(j)} = Z^j$, $j \geq 1$, $\beta(0) = 1$ and $\beta(m)$, $m \geq 1$ be*

$$\beta^{(m)} = \sum_{p=1}^m \frac{-1}{p} \sum_{v_1+\cdots+v_p=m, v_i \geq 1, k_1+\cdots+k_p=p-1, k_j \geq 0} \mathrm{trace}\left(T^{(v_1)}Z^{(k_1)}\ldots T^{(v_p)}Z^{(k_p)}\right),$$

*Then we have the following expansion on $\beta(t)$,*

$$\beta(t) = \sum_{m=0}^\infty \beta^{(m)}t^m, \ |t| < t_0.$$

Follow the same line as in (Lezaud, 1998) (Page 854-856), denote $\sigma^2 = \|f\|_\pi^2$ and $c = c_0 \geq \|\|D\|\|_\pi$ (defined in Lemma 8), then we have the following upper bound of $\beta(t)$.

$$\beta(t) = \beta^{(0)} + \beta^{(1)}t + \sum_{m=2} \beta^{(m)}t^m$$

$$\leq 1 + 0 + \sum_{m=2}^\infty \frac{\pi(f^m)t^m}{m!} + \sum_{m=2}^\infty \frac{\sigma^2\lambda t}{5c}\left(\frac{5ct}{1-\lambda_r}\right)^{m-1}$$

$$\leq \exp\left(\sum_{m=2}^\infty \frac{\pi(f^m)t^m}{m!} + \sum_{m=2}^\infty \frac{\sigma^2\lambda t}{5c}\left(\frac{5ct}{1-\lambda_r}\right)^{m-1}\right)$$

$$\leq \exp\left(\frac{\sigma^2}{c^2}\left(e^{tc} - 1 - tc\right) + \frac{\sigma^2\lambda t^2}{1-\lambda_r-5ct}\right)$$

$$:= \exp(g_1(t) + g_2(t)) \tag{18}$$

Now we are ready to derive the bound for the term $\mathbf{E}\left[e^{t\sum_i^n f_i(X_i)}\right]$. Following the results in Lemma 6, we consider a sequence of $f_k$ such that,

$$\left\|\left\|\left|E^{tf/2}\widehat{P}_\lambda E^{tf/2}\right\|\right\|\right\|_\pi = \lim_{k\to\infty} \left\|\left\|\left|E^{t\widehat{f}_k/2}\widehat{P}_\lambda E^{t\widehat{f}_k/2}\right\|\right\|\right\|_\pi.$$

Next, we construct the finite state space counterpart of each pair of $E^{t\widehat{f}_k/2}\widehat{P}_\lambda E^{t\widehat{f}_k/2}$ and $\pi$ by Lemma 7, i.e.,

$$\left\|\left\|\left|E^{t\widehat{f}_k/2}\widehat{P}_\lambda E^{t\widehat{f}_k/2}\right\|\right\|\right\|_\pi := \left\|\left\|\left|E^{t\mathcal{Y}_k/2}\widehat{Q}_\lambda E^{t\mathcal{Y}_k/2}\right\|\right\|\right\|_{\mu_k}$$

Let the random variable in the state space $\mathcal{Y}_k$ be $Y_k$, then the mean and variance of $Y_k$ is $\sum_{y\in\mathcal{Y}_k} \pi\left(\left\{x : \widehat{f}_k(x) = \mathcal{Y}\right\}\right) y = \pi\left(\widehat{f}_k\right) = 0$ and $\sum_{y\in\mathcal{Y}_k} \pi\left(\left\{x : \widehat{f}_k(x) = y\right\}\right) y^2 = \pi\left(\widehat{f}_k^2\right)$.

For each $k$, applying Eqn. (18) gives us,

$$\left\|\left\|\left|E^{t\mathcal{Y}_k/2}\widehat{Q}_\lambda E^{t\mathcal{Y}_k/2}\right\|\right\|\right\|_{\mu_k} \leq \exp\left(\frac{\pi\left(\widehat{f}_k^2\right)}{c^2}\left(e^{tc} - 1 - tc\right) + \frac{\pi\left(\widehat{f}_k^2\right)\lambda t^2}{1-\lambda_r-5ct}\right)$$

Note that as $k \to \infty$, we have $\pi\left(\widehat{f}_k^2\right) \to \pi(f^2) = \sigma^2$. Then we have the upper bound for each operator $\left\|\left\|E^{tf_i/2} P E^{tf_i/2}\right\|\right\|_\pi$, i.e., for any $\lambda \in [0, 1)$,

$$\left\|\left\|E^{tf/2} P_\lambda E^{tf/2}\right\|\right\|_\pi \leq \exp(g_1(t) + g_2(t))$$

where $g_1$ and $g_2$ are defined in Eqn. (18).

Consequently, we obtain the upper bound for $\mathbf{E}\left[e^{t \sum_i^n f_i(X_i)}\right]$ as follows, $\mathbf{E}\left[e^{t \sum_i^n f_i(X_i)}\right]$,

$$\mathbf{E}_\pi\left[e^{t \sum_{i=1}^n f_i(X_i)}\right] \leq \exp\left(\frac{n\sigma^2}{c^2}\left(e^{tc} - 1 - tc\right) + \frac{n\sigma^2 \max\{\lambda_r, 0\} t^2}{1 - \max\{\lambda_r, 0\} - 5ct}\right)$$

**Step 2** Use the convex analysis argument to derive the Bernstein's Inequality.

We first restate the following lemma (Lemma 9, (Jiang et al., 2018)) on the terms $g_1$ and $g_2$.

**Lemma 9.** *For $\lambda \in [0, 1)$, let $g_1(t) = \frac{n\sigma^2}{c^2}\left(e^{tc} - 1 - tc\right)$ and $g_2(t) = \frac{n\sigma^2 \max\{\lambda_r, 0\} t^2}{1 - \max\{\lambda_r, 0\} - 5ct}$, then for any $0 \leq t < (1 - \gamma)/5c$, the Frechet conjugates $(g_1 + g_2)^*$ satisfy the following inequalities.*

$$\text{if } \lambda \in (0, 1): \quad (g_1 + g_2)^*(\epsilon) := \sup_{0 \leq t < (1-\lambda)/5c} \{t\epsilon - g_1(t) - g_2(t)\} \geq \frac{\epsilon^2}{2}\left(\frac{1+\lambda}{1-\lambda}\sigma^2 + \frac{5c\epsilon}{1-\lambda}\right)^{-1}$$

$$\text{if } \lambda = 0: \quad (g_1 + g_2)^*(\epsilon) = g_1^*(\epsilon) \geq \frac{\epsilon^2}{2}\left(\sigma^2 + \frac{c\epsilon}{3}\right)^{-1}.$$

By the Chernoff bound, we have,

$$-\log \mathbf{P}\left(\frac{1}{n}\sum_{i=1}^n f_i(X_i) > \epsilon\right) \geq n \times \sup_{t \in \mathbb{R}}\{t\epsilon - g_1(t) - g_2(t)\}$$

Notice that $g_1(t) = O(t^2)$ and $g_2(t) = O(t^2)$ as $t \to 0$, then for some $t > 0$, we have $t\epsilon - g_1(t) - g_2(t) > 0$. Meanwhile, when $t \leq 0$, we have $t\epsilon - g_1(t) - g_2(t) \leq 0$. Thus, we can obtain that,

$$\sup\{t\epsilon - g_1(t) - g_2(t) : t > 0\} = \sup\{t\epsilon - g_1(t) - g_2(t) : t \in \mathbb{R}\} = (g_1 + g_2)^*(\epsilon).$$

Letting $\lambda = \max\{\lambda_r, 0\}$, $\alpha_1(\lambda) = (1 + \lambda)/(1 - \lambda)$, $\alpha_2(\lambda) = 5/(1 - \lambda)$ and $\alpha_2(0) = 1/3$ yields,

$$\mathbf{P}_\pi\left(\frac{1}{n}\sum_{i=1}^n f(X_i) > \epsilon\right) \leq \exp\left(-\frac{n\epsilon^2/2}{\alpha_1(\max\{\lambda_r, 0\}) \cdot \sigma^2 + \alpha_2(\max\{\lambda_r, 0\}) \cdot c\epsilon}\right). \quad (19)$$

This concludes the proof.

$\square$

## C    PROOF OF PROPOSITION 1

Let $\bar{\omega}_{t+1} = \Gamma_R(\tilde{\omega}_{t+1})$, and assume $\|\phi(s, a)\| \leq 1$ uniformly (see Assumption 1). Based on the approach in Appendix G.1 (Fu et al., 2020), it suffices to upper bound $\|\omega_{t+1} - \tilde{\omega}_{t+1}\|_2$. Observe that

$$\|\omega_{t+1} - \bar{\omega}_{t+1}\|_2 \leq \|\widehat{\Phi}\widehat{v} - \Phi v\|_2 \leq \|\Phi\|_2 \cdot \|\widehat{v} - v\|_2 + \|\widehat{\Phi} - \Phi\|_2 \cdot \|\widehat{v}\|_2,$$

where $\Phi$ and $v$ are given as follows:

$$\widehat{\Phi} = \left(\frac{1}{N}\sum_{l=1}^N \phi(s_l, a_l)\phi(s_l, a_l)^\top\right)^{-1},$$

$$\Phi = \left(\mathbf{E}_{\rho_{t+1}}\left[\phi(s, a)\phi(s, a)^\top\right]\right)^{-1},$$

$$\widehat{v} = \frac{1}{N}\sum_{l=1}^N \left((1 - \gamma)\sum_{i=0}^{m-1}\gamma^i r_{l,i} + \gamma^m Q_{\omega_t}(s_{l,m}, a_{l,m})\right) \cdot \phi(s_{l,m}, a_{l,m}),$$

$$v = \mathbf{E}_{\rho_{t+1}}\left[(1 - \gamma)\sum_{i=0}^{m-1}\left(\gamma^i r_{l,i} + \gamma^m \mathbf{P}_{\pi_{\theta_{t+1}}} Q_{\omega_t}(s_m, a_m)\right) \cdot \phi(s_m, a_m)\right].$$

Recall that the following assumptions are in place: (1) Spectral norm $\|\phi(s,a)\|_2 \le 1$, $\phi(s,a) \in \mathbb{R}^d$; (2) $|r(s,a)| \le r_{\max}$ and $\bar{r} = \mathbf{E}_{s,a} r(s,a)$; (3) $\|\omega_t\|_2 \le R$ and (4) the minimum singular value of the matrix $\mathbf{E}_{\rho_t}[\phi(s,a)\phi(s,a)^\top]$, $t \ge 1$ is uniformly lower bounded by $\sigma^*$. It can be shown that $\|\Phi\|_2 \le \frac{1}{\sigma^*}$.

Next, we derive the bound by appealing to Bernstein's Inequality with General Makovian samples. Following Theorem 2 (Jiang et al., 2018) (The proof of Bernstein's Inequality can be found in Appendix B), let $\pi_r$ be the invariant distribution (which is relevant to the current policy $\pi_k$) of the stationary Markov chain $\{r_t\}_{t=1}^m$. Suppose that it has non-zero right spectral gap $1 - \lambda_r > 0$. Let $\sigma_r^2 = \int (r - \bar{r})^2 \pi_r(dr)$. Then, we have that for any $\epsilon > 0$:

$$\mathbf{P}_{\pi_r}\left( \frac{1}{m} \sum_{i=1}^m (r_i - \bar{r}) > \epsilon \right) \le \exp\left( -\frac{m\epsilon^2/2}{\alpha_1(\max\{\lambda_r, 0\}) \cdot \sigma^2 + \alpha_2(\max\{\lambda_r, 0\}) \cdot r_{\max}\epsilon} \right),$$

where $\alpha_1(\lambda) = \frac{1+\lambda}{1-\lambda}$, $\quad \alpha_2(\lambda) = \begin{cases} \frac{1}{3} & \text{if } \lambda = 0 \\ \frac{5}{1-\lambda} & \text{if } \lambda \in (0,1) \end{cases}$.

We conclude that with probability at least $1 - p$,

$$\sum_{i=0}^{m-1} r_i \le \frac{\sqrt{\alpha_2^2(\max\{\lambda_r,0\})^2 \ln p^2 - 2m\alpha_1(\max\{\lambda_r,0\}) \ln p} - \alpha_2(\max\{\lambda_r,0\}) \ln p}{m} + \bar{r} := \tilde{r}_m.$$

It follows that with probability at least $1 - p$,
$$\|\hat{v}\|_2 \le (1-\gamma)\tilde{r}_m + \gamma^m R,$$

Further, note that
$$\|v\|_2 \le (1-\gamma)\bar{r} + \gamma^m R,$$

Since the minimum singular value of $\hat{\Phi}^{-1}$ is no less than $\frac{\sigma^*}{2}$ w.h.p. when $N$ is large enough, we have that
$$\|\hat{\Phi}\|_2 \le \frac{2}{\sigma^*}.$$

For convenience, define
$$\hat{X} \triangleq \left( \frac{1}{N} \sum_{l=1}^N \phi(s_l, a_l) \phi(s_l, a_l)^\top \right), \quad X \triangleq \left( \mathbf{E}_{\rho_{t+1}}[\phi(s,a)\phi(s,a)^\top] \right),$$

and define
$$Z \triangleq \hat{X} - X = \sum_{k=1}^N S_k, \tag{20}$$

$$S_k \triangleq \frac{1}{N}(\phi_k \phi_k^\top - X), \tag{21}$$

where $S_k, k = 1, \cdots, N$ are independent.

Next, we derive the uniform bound on the spectral norm of each summand as follows:
$$\|S_k\|_2 = \frac{1}{N}\|\phi_k\phi_k^\top - X\| \le \frac{1}{N}(\|\phi_k\phi_k^\top\| + \|X\|) \le \frac{2}{N}.$$

To this end, we bound the matrix variance statistic $V(Z)$:
$$V(Z) := \|\mathbf{E}[Z^2]\| = \|\sum_{k=1}^N \mathbf{E}[S_k^2]\|.$$

Note that the variance of each summand is given by

$$
\begin{aligned}
\mathbf{E}[S_k^2] =& \frac{1}{N^2} \mathbf{E}[(\phi_k \phi_k^\top - X)^2] \\
=& \frac{1}{N^2} \mathbf{E}[\|\phi_k\|^2 \cdot \phi\phi^\top - \phi\phi^\top X - X\phi\phi^\top + X^2] \\
\preccurlyeq& \frac{1}{N^2} [\mathbf{E}[\phi\phi^\top] - X^2] \\
\preccurlyeq& \frac{1}{N^2} X.
\end{aligned}
$$

Combining the above, we conclude that

$$
0 \preccurlyeq \sum_{k=1}^N \mathbf{E}[S_k^2] \preccurlyeq \frac{1}{N} X.
$$

Observe that

$$
\|X\| = \|\mathbf{E}[\phi\phi^\top]\|_2 \leq \mathbf{E}[\|\phi\phi^\top\|] = \mathbf{E}\|\phi\|^2 \leq 1.
$$

Since the spectral norm is the variance statistic given by

$$
V(Z) \leq \frac{1}{N} \|X\|,
$$

appealing to Bernstein's Inequality, we have that

$$
\mathbf{P}\{\|Z\| \geq t\} \leq 2d \exp\left( \frac{\frac{-t^2}{2}}{\frac{1}{N}\|X\| + \frac{2t}{3N}} \right),
$$

$$
\mathbf{E}[\|Z\|] \leq \sqrt{\frac{2}{N} \|X\| \log(2d)} + \frac{2}{3N} \log(2d)
$$

$$
\leq \sqrt{\frac{2}{N} \log(2d)} + \frac{2}{3N} \log(2d).
$$

This is to say, with probability at least $1 - p/2$, the following holds:

$$
\|X - \hat{X}\| \leq -\frac{2}{3N} \log \frac{p}{4d} + \sqrt{\frac{4}{9N^2} \log^2 \frac{p}{4d} - \frac{2}{N} \log \frac{p}{4d}}.
$$

In a nutshell, we have that

$$
\begin{aligned}
\|\widehat{\Phi} - \Phi\|_2 =& \|\hat{X}^{-1} - X^{-1}\|_2 \\
=& \|\hat{X}^{-1}(\hat{X} - X)X^{-1}\|_2 \\
=& \|\hat{\Phi}(\hat{X} - X)\Phi\|_2 \\
\leq& \frac{2}{(\sigma^*)^2} \|\hat{X} - X\|_2 \\
\leq& \frac{4}{\sqrt{N}(\sigma^*)^2} \cdot \left( -\frac{2}{3N} \log \frac{p}{4d} + \sqrt{\frac{4}{9N^2} \log^2 \frac{p}{4d} - \frac{2}{N} \log \frac{p}{4d}} \right).
\end{aligned}
$$

Similarly, the following inequality holds with probability at least $1 - p/2$:

$$
\|\widehat{v} - v\|_2 \leq -\frac{\delta_1}{3} \log \frac{p}{2(d+1)} + \sqrt{\frac{\delta_1^2}{9} \log^2 \frac{p}{2(d+1)} - 2\delta_2 \log \frac{p}{2(d+1)}},
$$

where $d$ is the dimension of vector $\varphi$, $\delta_1 = \frac{1}{N}((1-\gamma)(\tilde{r}_m + \bar{r}) + 2\gamma^m R)$ and $\delta_2 = \|\mathbf{E}[\hat{v} - v]\|_2$ satisfying

$$
\begin{aligned}
\delta_2 \leq& \frac{1}{N} [(1-\gamma)(|\tilde{r}_m|(|\tilde{r}_m - \bar{r}| + \gamma^m R|\tilde{r}_m - \bar{r}|))] \\
\leq& \frac{1-\gamma}{N} [r_{\max} + \gamma^m R]|\tilde{r}_m - \bar{r}|.
\end{aligned}
$$

Summarizing, we have that

$$\|\omega_{t+1} - \bar{\omega}_{t+1}\|_2 \leq \|\Phi\|_2 \cdot \|\hat{v} - v\|_2 + \|\hat{\Phi} - \Phi\|_2 \cdot \|\hat{v}\|_2$$

$$\leq -\frac{\delta_1}{3\sigma^*} \log \frac{p}{2(d+1)} + \sqrt{\frac{\delta_1^2}{9} \log^2 \frac{p}{2(d+1)} - 2\delta_2 \log \frac{p}{2(d+1)}}$$

$$+ \frac{4((1-\gamma)\tilde{r}_m + \gamma^m R)}{\sqrt{N}(\sigma^*)^2} \left( -\frac{2}{3N} \log \frac{p}{4d} + \sqrt{\frac{4}{9N^2} \log^2 \frac{p}{4d} - \frac{2}{N} \log \frac{p}{4d}} \right),$$

which indicates that with probability at least $1 - p$,

$$|Q_{\omega_{t+1}} - Q_{\bar{\omega}_{t+1}}| \leq \left( \frac{4((1-\gamma)\tilde{r}_m + \gamma^m R)}{\sqrt{N}(\sigma^*)^2} \left( -\frac{2}{3N} \log \frac{p}{4d} + \sqrt{\frac{4}{9N^2} \log^2 \frac{p}{4d} - \frac{2}{N} \log \frac{p}{4d}} \right) \right)$$

$$\triangleq \epsilon_Q. \tag{22}$$

**Remark.** In the case when Assumption 1 does not hold, i.e., we have

$$\inf_{\bar{\omega} \in \Omega} \mathbf{E}_{\rho^{\pi_\theta}} \left[ \left( (T^{\pi_\theta})^m Q_\omega - \bar{\omega}^\top \phi \right)(s, a) \right] = c_1,$$

where $c_1 > 0$ is a constant. Let $\bar{\omega}_{t+1} = \Gamma_R(\tilde{\omega}_{t+1})$, recall that $\tilde{\omega}$ denotes the solution of Eqn. (8) and $\omega$ denotes the sample-based solution, then we have

$$|Q_{\bar{\omega}_{t+1}} - Q_{\tilde{\omega}_{t+1}}| = c_1$$

From Eqn. (22), we obtain that,

$$|Q_{\omega_{t+1}} - Q_{\bar{\omega}_{t+1}}| \leq \epsilon_Q$$

Then the difference between the sample-based solution and the underlying true solution of Eqn. (8) is,

$$|Q_{\omega_{t+1}} - Q_{\tilde{\omega}_{t+1}}| \leq \epsilon_Q + c_1.$$

Note that when Assumption 1 holds,

$$Q_{\tilde{\omega}_{t+1}} = Q_{\bar{\omega}_{t+1}}.$$

## D PROOF OF LEMMA 1

Recall $\beta = f_{k,t} + C_{k,t} - \mathbf{E}[C_{k,t}] - \mathbf{E}[f_{k,t}]$. We also have the following definitions:

$$C_{k,t,1} \triangleq 1/l \sum_{i=1}^{l} (Q_{\omega_{t+1}}(s_i, a_i) \nabla_\theta \pi_{\theta_k}(a_i|s_i) - Q_{\tilde{\omega}_{t+1}}(s_i, a_i) \nabla_\theta \pi_{\theta_k}(a_i|s_i)),$$

$$C_{k,t,2} \triangleq 1/l \sum_{i=1}^{l} (Q_{\tilde{\omega}_{t+1}}(s_i, a_i) \nabla_\theta \pi_{\theta_k}(a_i|s_i) - Q^{\pi_{\theta_t}}(s_i, a_i) \nabla_\theta \pi_{\theta_k}(a_i|s_i)),$$

$$f_{k,t} \triangleq 1/l \sum_{i=1}^{l} Q^{\pi_{\theta_t}}(s_i, a_i) \nabla_\theta \pi_{\theta_k}(a_i|s_i),$$

$$C_{k,t} \triangleq C_{k,t,1} + C_{k,t,2}.$$

Next we evaluate $\mathbf{E}[\|f_{k,t} + C_{k,t} - \mathbf{E}[C_{k,t}] - \mathbf{E}[f_{k,t}]\|^2]$ as follows:

$$\|f_{k,t} + C_{k,t} - \mathbf{E}[C_{k,t}] - \mathbf{E}[f_{k,t}]\|^2$$

$$= (f_{k,t} + C_{k,t})(f_{k,t} + C_{k,t})^\top + (\mathbf{E}[C_{k,t}] + \mathbf{E}[f_{k,t}])(\mathbf{E}[C_{k,t}] + \mathbf{E}[f_{k,t}])^\top$$

$$- 2(f_{k,t} + C_{k,t})(\mathbf{E}[C_{k,t}] + \mathbf{E}[f_{k,t}])^\top$$

$$\leq (f_{k,t} + C_{k,t})(f_{k,t} + C_{k,t})^\top + (\mathbf{E}[C_{k,t}] + \mathbf{E}[f_{k,t}])(\mathbf{E}[C_{k,t}] + \mathbf{E}[f_{k,t}])^\top. \tag{23}$$

Note that $C_{k,t}$ and $f_{k,t}$ are both bounded above since $Q$-function is bounded and $\nabla_\theta \pi_\theta(a|s)$ is bounded (see Assumption 4), i.e.,

$$\|\nabla \pi(a|s)\| \leq C_\psi,$$

$$\|Q(s,a)\| \leq \sum_{t=1}^{\infty} \gamma^t r_{\max} = \frac{r_{\max}}{1-\gamma}.$$

Then we have the following bounds for $C_{k,t}$ and $f_{k,t}$:

$$\|C_{k,t}\| \leq 2C_\psi \frac{r_{\max}}{1-\gamma},$$

$$\|f_{k,t}\| \leq C_\psi \frac{r_{\max}}{1-\gamma}.$$

Taking expectation over both sides of the inequality (23), we have that

$$\mathbf{E}[\|\beta\|^2] \leq 1 \cdot \|\mathbf{E}[C_{k,t}] + \mathbf{E}[f_{k,t}]\|^2 + \mathbf{E}[(f_{k,t} + C_{k,t})(f_{k,t} + C_{k,t})^\top].$$

Let $M_n = 1$ and $\sigma^2 = \mathbf{E}[(f_{k,t} + C_{k,t})(f_{k,t} + C_{k,t})^\top]$. Then we have that

$$\mathbf{E}[\|\beta\|^2] \leq M_n \cdot \|\mathbf{E}[C_{k,t}]\| + \sigma^2,$$

where $\sigma$ depends on probability $p$ as indicated in Eqn. (22).

## E    PROOF OF LEMMA 2

Recall that $b = \mathbf{E}[C_{k,t}]$ and

$$C_{k,t} := C_{k,t,1} + C_{k,t,2}$$

$$= 1/l \sum_{i=1}^{l} \Big( Q_{\omega_{t+1}}(s_i, a_i) \nabla_\theta \pi_{\theta_k}(a_i|s_i) - Q_{\widetilde{\omega}_{t+1}}(s_i, a_i) \nabla_\theta \pi_{\theta_k}(a_i|s_i) +$$

$$(Q_{\widetilde{\omega}_{t+1}}(s_i, a_i) \nabla_\theta \pi_{\theta_k}(a_i|s_i) - Q^{\pi_{\theta_t}}(s_i, a_i) \nabla_\theta \pi_{\theta_k}(a_i|s_i) \Big).$$

Next, we evaluate $\|b\|^2$. Observe that (see also Appendix D)

$$\|C_{k,t}\| \leq 2C_\psi \frac{r_{\max}}{1-\gamma}.$$

Let $\zeta = 2\|C_\psi \frac{r_{\max}}{1-\gamma}\|$. Then we have

$$\|b\|^2 = \|\mathbf{E}[C_{k,t}]\|^2 \leq \mathbf{E}[\|C_{k,t}\|^2] \leq \zeta^2.$$

## F    PROOF OF LEMMA 3 AND LEMMA 4

- [Lemma 3] Given Critic parameter $\omega$ in the objective function, it can be seen that $\|\nabla h(\omega, \theta) - \nabla h(\omega, \theta')\| \leq Q_{\max} \|\nabla \pi_\theta - \nabla \pi_{\theta'}\|$. Since value function is bounded (e.g., $Q_{\max}$) and the score function $\nabla \pi_\theta$ is $L_\psi$-smooth (ref. Assumption 6), the constant in Assumption 4 can be easily determined by $L = Q_{\max} L_\psi$.

- [Lemma 4] Since the objective function is finite, let $h_{\max} = \max_{\theta \neq \theta^*} h(\theta, \omega)$, $h^*_{\max} = \max_{\theta = \theta^*} h(\theta, \omega)$,. In the case when the gradient is non-zero, let $g_{\min} = \min_{\theta \neq \theta^*} \nabla h$, then we can determine $\mu = \frac{g_{\min}}{h^*_{\max} - h_{\max}} \geq 0$.

## G    PROOF OF PROPOSITION 2

Observe that the Actor updates use the biased stochastic gradient methods (SGD). For simplicity, we adopt the following notations to study the Actor update:

$$\theta_{k+1} = \theta_k + \alpha(\nabla h(\omega, \theta_k) + b(t) + \beta(t)). \tag{24}$$

where $b(t) = \mathbf{E}[C_{k,t}]$ is the bias, $\alpha$ is the step size, and

$$\beta = f_{k,t} + C_{k,t} - \mathbf{E}[C_{k,t}] - \mathbf{E}[f_{k,t}]$$

is the zero-mean noise. Note that the objective function $h(\omega, \theta_k)$ is a function of $\theta$. Denote the optimal value (in this iteration of the Actor update) by $h(\omega, \theta^*)$.

We prove the following lemma on the modified version of the descent lemma for smooth function (cf. (Ajalloeian & Stich, 2020; Nesterov, 2003)).

**Lemma 10.** *Suppose Assumption 3 and 4 hold. Then, for any stepsize $\alpha \leq \frac{1}{(M_n+1)L}$, the following inequality holds with probability at least $1 - p$:*

$$\mathbf{E}[h(\omega, \theta_{k+1}) - h(\omega, \theta_k)|\theta_k] \leq \frac{\alpha}{2}\zeta^2 + \frac{\alpha^2 L}{2}\sigma^2 - \frac{\alpha}{2}\|\nabla h(\omega, \theta_k)\|^2.$$

Observe that under the PL-condition (Assumption 4), we have the following recursion:

$$\mathbf{E}[h(\omega, \theta_{k+1}) - h(\omega, \theta^*)] \leq (1 - \alpha\mu)\mathbf{E}[h(\omega, \theta_k) - h(\omega, \theta^*)] + \frac{\alpha}{2}\zeta^2 + \frac{\alpha^2 L}{2}\sigma^2. \tag{25}$$

By applying Eqn. (25) recursively, we obtain the desired results in Proposition 2.

## H    PROOF OF PROPOSITION 3

We first prove the following lemma on the relation between Actor parameter $\theta$ and the objective function $h(\omega, \theta)$.

**Lemma 11.** *There exist a contant $L_h > 0$ and an open ball $S_\epsilon(\theta_t^*)$ such that for any $\theta_t \in B_\epsilon(\theta_t^*)$ the following holds for any $t > 0$.*
$$\mathbf{E}[\|\pi_{\theta_t} - \pi^*\|_{\mathrm{TV}}] \leq L_h \mathbf{E}[h(\omega, \theta_t^*) - h(\omega, \theta_t)].$$

*Proof.* By Taylor's expansion, we have
$$h(\omega, \theta^*) = h(\omega, \theta_t) + \nabla h(\omega, \theta_t)(\theta_t^* - \theta_t) + o(\|\theta_t^* - \theta_t\|).$$
Since $h(\omega, \cdot)$ satisfies Polyak-Lojasiewicz Condition, it follows that
$$\|\nabla h(\omega, \theta)\| \geq 2\mu(h(\omega, \theta^*) - h(\omega, \theta)) := L_g \text{ for all } \theta.$$
Note that $L_g > 0$ when $\theta \neq \theta^*$. Then we have that
$$\begin{aligned}
h(\omega, \theta_t^*) - h(\omega, \theta_t) =&|\nabla h(\omega, \theta_t)(\theta^* - \theta_t) + o(\|\theta^* - \theta_t\|)| \\
\geq&|\nabla h(\omega, \theta_t)(\theta_t^* - \theta_t)| - |o(\|\theta^* - \theta_t\|)| \\
\geq&L_g\|\theta_t^* - \theta_t\| - L_o\|\theta_t^* - \theta_t\| \\
=&(L_g - L_o)\|\theta_t^* - \theta_t\|,
\end{aligned}$$
where the last inequality uses the fact that there exists $\epsilon$ such that when $\|\theta_t - \theta_t^*\| \leq \epsilon$,
$$|o(\|\theta_t^* - \theta_t\|)| \leq L_o\|\theta_t^* - \theta_t\|, \ \ L_o < L_g.$$

Taking expectation over both sides gives
$$\begin{aligned}
\mathbf{E}[h(\omega, \theta_t^*) - h(\omega, \theta_t)] =&(L_g - L_o)\mathbf{E}[\|\theta_t^* - \theta_t\|] \\
\geq&(L_g - L_o)\|\mathbf{E}[\theta_t^* - \theta_t]\|.
\end{aligned}$$

Then we conclude that the parameter of interest $L_h$,
$$L_h = \frac{C_\pi}{L_g - L_o} > 0.$$
where $C_\pi$ is defined in Assumption 4.                                    $\square$

We are ready to present the proof of Proposition 3. Based on the definition of $\mathcal{E}_{\hat{J},t}$ and $\mathcal{E}_{\hat{T},t}$, we derive the upper bound for each term respectively.
$$\begin{aligned}
\mathcal{E}_{\hat{J},t} =&(\boldsymbol{I} - \gamma\boldsymbol{P}_{\hat{\pi}_{t+1}})^{-1} - (\boldsymbol{I} - \gamma\boldsymbol{P}_{\widetilde{\pi}_{t+1}})^{-1} \\
=&(\boldsymbol{I} - \gamma\boldsymbol{P}_{\widetilde{\pi}_{t+1}})^{-1}\left(\gamma\boldsymbol{P}_{\widetilde{\pi}_{t+1}} - \gamma\boldsymbol{P}_{\hat{\pi}_{t+1}}\right)(\boldsymbol{I} - \gamma\boldsymbol{P}_{\hat{\pi}_{t+1}})^{-1}.
\end{aligned}$$

Observe that value function $v$ is smooth and upper bounded. We denote the smoothness parameter by $L_v$, the upper bound by $\|v\| \leq V^{\max}$, and the smoothness of the reward function by $L_r$.

By taking the norm of both sides and applying Assumption 3, 4 and 5, we obtain
$$\|\mathcal{E}_{\hat{J},t}\| \leq M^2 L_J L_v\|\widetilde{\pi}_{t+1} - \hat{\pi}_{t+1}\|_{\mathrm{TV}}.$$

Further, observe that

$$\mathcal{E}_{\hat{T},t} = \boldsymbol{r}_{\hat{\pi}_{t+1}} + \gamma \boldsymbol{P}_{\hat{\pi}_{t+1}} \boldsymbol{v}^{\hat{\pi}_t} - (\boldsymbol{r}_{\widetilde{\pi}_{t+1}} + \gamma \boldsymbol{P}_{\widetilde{\pi}_{t+1}} \boldsymbol{v}^{\hat{\pi}_t}),$$
$$= \boldsymbol{r}_{\hat{\pi}_{t+1}} - \boldsymbol{r}_{\widetilde{\pi}_{t+1}} + \gamma (\boldsymbol{P}_{\hat{\pi}_{t+1}} - \boldsymbol{P}_{\widetilde{\pi}_{t+1}}) \boldsymbol{v}^{\hat{\pi}_t}.$$

By taking the norm of both sides and applying Assumption 5, we obtain

$$\|\mathcal{E}_{\hat{T},t}\| = \|\boldsymbol{r}_{\hat{\pi}_{t+1}} - \boldsymbol{r}_{\widetilde{\pi}_{t+1}}\| + \|\gamma (\boldsymbol{P}_{\hat{\pi}_{t+1}} - \boldsymbol{P}_{\widetilde{\pi}_{t+1}}) \boldsymbol{v}^{\hat{\pi}_t}\|$$
$$\leq (L_r + \gamma V^{\max})\|\widetilde{\pi}_{t+1} - \hat{\pi}_{t+1}\|_{\mathrm{TV}}$$
$$:= L_T^{\max}.$$

Recall the definition of $\mathcal{E}_t$ is given as

$$\mathcal{E}_t = -\left( \mathcal{E}_{\hat{J},t}(\boldsymbol{v}^{\hat{\pi}_t} - T(\boldsymbol{v}^{\hat{\pi}_t})) + \boldsymbol{J}_{\hat{\boldsymbol{v}}_t}^{-1} \mathcal{E}_{\hat{T},t} + \mathcal{E}_{\hat{T},t} \mathcal{E}_{\hat{J},t} \right).$$

Taking the norm and expectation on both sides yields that

$$\|\mathbf{E}[\mathcal{E}_t]\| \leq \mathbf{E}[\|\mathcal{E}_t\|] = \mathbf{E}\left[ \|\mathcal{E}_{\hat{J},t}(\boldsymbol{v}^{\hat{\pi}_t} - T(\boldsymbol{v}^{\hat{\pi}_t})) + \boldsymbol{J}_{\hat{\boldsymbol{v}}_t}^{-1} \mathcal{E}_{\hat{T},t} + \mathcal{E}_{\hat{T},t} \mathcal{E}_{\hat{J},t}\| \right]$$
$$\leq L_{\mathcal{E}} \mathbf{E}[\|\widetilde{\pi}_{t+1} - \hat{\pi}_{t+1}\|_{\mathrm{TV}}],$$

where $L_{\mathcal{E}} = (2V^{\max}K + L_T^{\max})M^2 L_v L_J + M(L_r + \gamma V^{\max}) > 0$ is a constant. Since $\widetilde{\pi}_{t+1} = \pi_{t+1}^*$ is the greedy solution, we thereby complete the proof of Proposition 3.

# I PROOF OF THEOREM 1

Following the value function update rule, we have

$$\boldsymbol{v}^{\hat{\pi}_{t+1}} = \boldsymbol{v}^{\hat{\pi}_t} - \left( \boldsymbol{J}_{\hat{\boldsymbol{v}}_t}^{-1} \left( \boldsymbol{v}^{\hat{\pi}_t} - T(\boldsymbol{v}^{\hat{\pi}_t}) \right) + \mathcal{E}_t \right)$$
$$= \boldsymbol{v}^{\hat{\pi}_t} - (L(t) + \mathcal{E}_t)$$
$$:= \boldsymbol{v}^{\hat{\pi}_t} - \hat{L}(t).$$

Then, the difference between $\boldsymbol{v}^{\hat{\pi}_{t+1}}$ and $\boldsymbol{v}^*$ is given by

$$\boldsymbol{v}^* - \boldsymbol{v}^{\hat{\pi}_{t+1}} = \boldsymbol{v}^* - \boldsymbol{v}^{\hat{\pi}_t} + \boldsymbol{J}_{\hat{\boldsymbol{v}}_t}^{-1} \left( \boldsymbol{v}^{\hat{\pi}_t} - T(\boldsymbol{v}^{\hat{\pi}_t}) \right) + \mathcal{E}_t. \qquad (26)$$

Observe the following result holds for any $\hat{\pi}_t$,

$$(\boldsymbol{v}^{\hat{\pi}_t} - T(\boldsymbol{v}^{\hat{\pi}_t})) - \underbrace{(\boldsymbol{v}^* - T(\boldsymbol{v}^*))}_{=0} \geq \boldsymbol{J}_{\hat{\boldsymbol{v}}_t}^2 (\boldsymbol{v}^{\hat{\pi}_t} - \boldsymbol{v}^*). \qquad (27)$$

Recall our decomposition of the value function update is given as

$$\hat{L}(t) = L(t) + \underbrace{\hat{L}(t) - \mathbf{E}[\hat{L}(t)]}_{\text{Martingale Difference Noise: } \mathcal{N}(t)} + \underbrace{\mathbf{E}[\hat{L}(t)] - L(t)}_{\text{Bias: } \mathcal{B}(t)}.$$

Plugging Eqn. (27) into Eqn. (26), we obtain

$$\boldsymbol{v}^* - \boldsymbol{v}^{\hat{\pi}_{t+1}} = \boldsymbol{v}^* - \boldsymbol{v}^{\hat{\pi}_t} + \left( \boldsymbol{J}_{\hat{\boldsymbol{v}}_t}^{-1} \left( \boldsymbol{v}^{\hat{\pi}_t} - T(\boldsymbol{v}^{\hat{\pi}_t}) \right) + \mathcal{E}_t \right)$$
$$\geq (\boldsymbol{I} - \boldsymbol{J}_{\boldsymbol{v}^{\hat{\pi}_t}}))(\boldsymbol{v}^* - \boldsymbol{v}^{\hat{\pi}_t}) + \mathcal{B}(t) + \mathcal{N}(t)$$
$$= \gamma \boldsymbol{P}_{\widetilde{\pi}_{t+1}}(\boldsymbol{v}^* - \boldsymbol{v}^{\hat{\pi}_t}) + \mathcal{B}(t) + \mathcal{N}(t).$$

Taking expectation on both sides yields that

$$\mathbf{E}[\boldsymbol{v}^* - \boldsymbol{v}^{\hat{\pi}_{t+1}}|\boldsymbol{v}^{\hat{\pi}_t}] \geq \gamma \boldsymbol{P}_{\widetilde{\pi}_{t+1}}(\boldsymbol{v}^* - \boldsymbol{v}^{\hat{\pi}_t}) + \mathcal{B}(t).$$

Applying the above inequality recursively gives that

$$
\begin{aligned}
\mathbf{E}\left[\boldsymbol{v}^* - \boldsymbol{v}^{\hat{\pi}_{t+1}}\right] \geq & \gamma^{t+1}\mathbf{E}\left[\left(\prod_{i=0}^{t}\boldsymbol{P}_{\widetilde{\pi}_{t+1-i}}\right)\right](\boldsymbol{v}^* - \boldsymbol{v}^{\pi_0}) \\
& + \sum_{i=1}^{t}\gamma^i\mathbf{E}\left[\left(\prod_{j=0}^{i-1}\boldsymbol{P}_{\widetilde{\pi}_{t+1-j}}\right)\right](\mathcal{B}(t-i)) + \mathcal{B}(t) \\
:= & \gamma^{t+1}\bar{\boldsymbol{P}}_{t+1}(\boldsymbol{v}^* - \boldsymbol{v}^{\pi_0}) + \sum_{i=1}^{t}\gamma^i\bar{\boldsymbol{P}}_i\mathcal{B}(t-i) + \mathcal{B}(t),
\end{aligned}
\tag{28}
$$

with $\bar{\boldsymbol{P}}_{t+1} = \mathbf{E}\left[\left(\prod_{i=0}^{t}\boldsymbol{P}_{\widetilde{\pi}_{t+1-i}}\right)\right]$. Taking norm on both sides of Eqn. (28) yields the desired results.

## J PROOF OF THEOREM 2

Based on the update rule of the value function, we have

$$
\begin{aligned}
\boldsymbol{v}^* - \boldsymbol{v}^{\hat{\pi}_{t+1}} = & \boldsymbol{J}_{\hat{\boldsymbol{v}}_t}^{-1}\boldsymbol{J}_{\hat{\boldsymbol{v}}_t}(\boldsymbol{v}^* - \boldsymbol{v}^{\hat{\pi}_t}) + \boldsymbol{J}_{\hat{\boldsymbol{v}}_t}^{-1}\left(\boldsymbol{v}^{\hat{\pi}_t} - T(\boldsymbol{v}^{\hat{\pi}_t})\right) + \mathcal{E}_t \\
\leq & \boldsymbol{J}_{\hat{\boldsymbol{v}}_t}^{-1}\boldsymbol{J}_{\hat{\boldsymbol{v}}_t}(\boldsymbol{v}^* - \boldsymbol{v}^{\hat{\pi}_t}) - \boldsymbol{J}_{\hat{\boldsymbol{v}}_t}^{-1}\boldsymbol{J}_{\boldsymbol{v}^*}(\boldsymbol{v}^* - \boldsymbol{v}^{\hat{\pi}_t}) - \mathcal{E}_t \\
\leq & \boldsymbol{J}_{\hat{\boldsymbol{v}}_t}^{-1}\left[\boldsymbol{J}_{\hat{\boldsymbol{v}}_t} - \boldsymbol{J}_{\boldsymbol{v}^*}\right](\boldsymbol{v}^* - \boldsymbol{v}^{\hat{\pi}_t}) + \mathcal{E}_t,
\end{aligned}
$$

which implies that

$$
\mathbf{E}[\boldsymbol{v}^* - \boldsymbol{v}^{\hat{\pi}_{t+1}}|\boldsymbol{v}^{\hat{\pi}_t}] \leq \boldsymbol{J}_{\hat{\boldsymbol{v}}_t}^{-1}\left[\boldsymbol{J}_{\hat{\boldsymbol{v}}_t} - \boldsymbol{J}_{\boldsymbol{v}^*}\right](\boldsymbol{v}^* - \boldsymbol{v}^{\hat{\pi}_t}) + \mathcal{B}(t).
$$

Then, taking norm on both sides of the inequality above gives

$$
\|\mathbf{E}[\boldsymbol{v}^* - \boldsymbol{v}^{\hat{\pi}_{t+1}}|\boldsymbol{v}^{\hat{\pi}_t}]\| \leq \|\boldsymbol{J}_{\hat{\boldsymbol{v}}_t}^{-1}\left[\boldsymbol{J}_{\hat{\boldsymbol{v}}_t} - \boldsymbol{J}_{\boldsymbol{v}^*}\right](\boldsymbol{v}^* - \boldsymbol{v}^{\hat{\pi}_t}) + \mathcal{B}(t)\|.
\tag{29}
$$

Let $H_t = \|\boldsymbol{J}_{\hat{\boldsymbol{v}}_t}^{-1}\left[\boldsymbol{J}_{\hat{\boldsymbol{v}}_t} - \boldsymbol{J}_{\boldsymbol{v}^*}\right]\|$. It follows from Assumption 5 that

$$
H_t \leq ML_J\|\boldsymbol{v}^{\hat{\pi}_t} - \boldsymbol{v}^*\|^q.
$$

where $L_J$ is defined in Assumption 5.

Hence, we have that

$$
\|\mathbf{E}[\boldsymbol{v}^{\hat{\pi}_{t+1}} - \boldsymbol{v}^*]\| \leq \|\boldsymbol{J}_{\hat{\boldsymbol{v}}_t}^{-1}\|\|\boldsymbol{J}_{\hat{\boldsymbol{v}}_t} - \boldsymbol{J}_{\boldsymbol{v}^*}\|\|\boldsymbol{v}^{\hat{\pi}_t} - \boldsymbol{v}^*\| + \|\mathcal{B}(t)\|.
\tag{30}
$$

By applying Eqn. (30) recursively, we conclude that

$$
\|\mathbf{E}[\boldsymbol{v}^{\hat{\pi}_{t+1}} - \boldsymbol{v}^*]\| \leq \left(\prod_{i=0}^{t}H_{t-i}\right)\|\mathbf{E}[\boldsymbol{v}^* - \boldsymbol{v}^{\pi_0}]\| + \sum_{i=1}^{t}\left(\prod_{j=1}^{i}H_{t-j}\right)\|\mathcal{B}(t-i)\| + \|\mathcal{B}(t)\|.
$$

## K EXPERIMENTS

**A Summary of Theoretical Results.** This work aims to provide a comprehensive answer to the question of "whether and under what conditions online learning (e.g., the general algorithm like AC) can be significantly accelerated by a warm-start policy from offline RL". Our key observations are as follows.

(1) Our results in Theorem 1 and Theorem 2 point out that the warm-start policy goes hand-by-hand with the approximation error to influence the learning performance (see the table below for the summary). The intricate relationship between the warm-start policy and the approximation error can be identified by studying the structure of the bounds in Theorem 4 and Theorem 5, where the warm-start policy not only has impact on the first term (e.g., $\boldsymbol{v}^* - \boldsymbol{v}^{\pi_0}$) but also the bias propagation through $H_0 := \|\boldsymbol{J}_{\hat{\boldsymbol{v}}_0}^{-1}\left[\boldsymbol{J}_{\hat{\boldsymbol{v}}_0} - \boldsymbol{J}_{\boldsymbol{v}^*}\right]\|$. Meanwhile, the biases have impact on the effect of the warm-start policy (the first term in the bounds) through $H_t$ directly.

(2) In Theorem 1, we point out that the bias terms have direct impact on "*whether the warm-start policy is able to facilitate the online learning*". For instance, "even when the warm-start

policy is nearly-optimal", there is still no guarantee that online fine-tuning can improve the policy much if there exist biases in the approximation errors in online Actor and Critic updates and these biases are not dealt with properly. To clarify further, consider the case when the bias is always positive, i.e., $\mathcal{B}(t) > 0$ for all $t \geq 0$, the lower bound is always positive and bounded away from zero.

(3) In Theorem 2, we aim to answer the question: "*under what conditions online learning can be significantly accelerated by a warm-start policy?*". Consider the case when the approximation error is unbiased (such that the A-C can be viewed as the Newton's Method). Clearly, we have $\|\mathbf{E}[\boldsymbol{v}^{\hat{\pi}_{t+1}} - \boldsymbol{v}^*]\| \leq \left(\prod_{i=0}^{t} H_{t-i}\right) \|\boldsymbol{v}^* - \boldsymbol{v}^{\pi_0}\|$, which can decrease quickly as long as the warm-start policy $\pi_0$ is not far away from the optimal policy $\pi^*$ and also satisfies Assumption 5. Intuitively, this result shows that "the imperfections of the warm-start policy can be 'washed out' by the (superlinear) Newton step" and corroborates with the observation in the very recent literature (Bertsekas, 2022b.). We remark that this phenomenon has not been formalized by previous works on the A-C algorithm.

| | $\mathcal{B}(t) \to 0$ | $\|\mathcal{B}(t)\| > 0$ |
|---|---|---|
| when the distance between $\pi_0$ and $\pi^*$ is small | The warm-start can facilitate the online convergence (Theorem 2) (Empirical studies:Silver et al. (2017; 2018)) | Biases can throttle the convergence significantly due to the accumulation effect (Theorem 1) (Empirical studies: Uchendu et al. (2022)) |
| when the distance between $\pi_0$ and $\pi^*$ is relatively large | The imperfections of the warm-start can be "washed out" by online learning (Theorem 2, Eqn. (30)) (Empirical studies: Bertsekas (2022b)) | The warm-start policy goes hand-by-hand with the approximation error to influence the learning performance |

**Empirical Results.** We consider experiments over the Gridworld benchmark task. In particular, we consider the following sizes of the grid to represent different problem complexity, i.e., $10 \times 10$, $15 \times 15$ and $20 \times 20$. The goal of the agent is to find a way (policy) to travel from a specified start location, e.g., the red square in Fig. 3, to an assigned target location, e.g., the red hexagram in Fig. 3, such that the (discounted) accumulative reward along the way is maximized. Specifically, the action space contains 4 discrete actions, namely, up, down, left, right, which are represented as 1,2,3,4 in the algorithm, respectively. The reward in the goal state is defined as 10 and in the bad state , e.g., the black cube in Fig. 3, is -6. The rest of the states result in the reward $-1$. The discounting factor is set as $\gamma = 0.9$. We consider the grid with 10 rows and 10 columns such that the state space has 100 states. The transition properties of the environment is as follows: the agent will transfer to next state following the chosen action with probability 0.7; the agent will go left of the desired action with probability 0.15 and go right with with probability 0.15. For each experiment, the shaded area represents a standard deviation of the average evaluation over 5 training seeds.

Specifically, we consider the following A-C algorithm to solve the Gridworld benchmark task,

Critic Update: The Critic updates its value by applying the Bellman evaluation operator $(T^\pi)$ for $m$-times $(m \geq 1)$, i.e., given policy $\pi$, at the $t$-th step A-C update,
$$\boldsymbol{v}(t+1) = (T^\pi)^m(\boldsymbol{v}(t)). \tag{31}$$

Actor Update: The Actor updates the policy by a greedy step to maximize the learnt $\boldsymbol{v}$ value, i.e.,
$$\pi' = \arg\max_\pi T^\pi(\boldsymbol{v}(t+1)). \tag{32}$$

**Impact of the Warm-Start Policy.** We first consider the impact of the Warm-Start policy in the ideal setting, where both the Critic update and Actor update is nearly accurate as in ADP. In this case, we let $m$ be large enough, e.g., $m = 1000$, in the Critic update Eqn. (31). As observed in Fig. 4, a 'good' Warm-Start policy can efficiently accelerate the learning process, e.g., it only takes two iterations to convergence with a Warm-Start policy. Meanwhile, in all three cases, the performance gap $\|\boldsymbol{v}(t) - \boldsymbol{v}^*\|$ decays over time which reflects our discovery in Theorem 2. Specifically, when the Warm-Start policy is not 'good' enough (or even no Warm-Start), the A-C algorithm can still be able to improve the learning performance overtime (see e.g., the first term on the right side of the upper bound in Theorem 2).

**Impact of the Approximation Error in the Critic Update.** We evaluate the impact of the approximation error in the Critic update on the convergence behavior by two approaches. (1) First, we study the Critic update with finite time Bellman evaluation, e.g., $m = 500, 50, 20, 5$. As shown in Fig. 5, the inaccurate Critic update impacts the convergence behavior as expected. The case when $m = 5$ shows that the finite time Bellman evaluation may contribute to the slower convergence. (2) Next, we consider the general case when there is approximation error in the Critic update. In particular, we add the uniform noise $e(t)$ in the value function with different bias, e.g.,$\mathbf{E}[e(t)] = 0, 0.5, 1, -1$. Meanwhile, we also consider the case when the bias can be either $+0.5$ or $-0.5$ in the learning process, e.g., $\mathbf{E}[e(t)] = 0.5$ with probability 0.5 and $\mathbf{E}[e(t)] = -0.5$ with probability 0.5. The resulting convergence behavior is presented in Fig. 6. Notably, it can be clearly seen that both the positive and negative bias may result in an error floor and 'prevent' the algorithm from converging to the optimal (e.g., the last two terms of the lower bound in Theorem (1)). The experiment results in Fig. 6 corroborate our theoretical findings in Theorems 1, 2 and 1.

**Impact of the Approximation Error in the Actor Update.** We investigate the learning performance of the A-C algorithm under inaccurate Actor update. In particular, we add the perturbation on the learnt policy in Eqn. (32) as follows,

$$\text{Policy}(s) = \begin{cases} \text{Policy}(s), & p, \\ \text{randi}([1, 4]), & 1 - p. \end{cases}$$

where $\text{Policy}(s)$ denotes the action should the agent take at the current state $s$ following the learnt policy and $\text{randi}([1, 4])$ is a random function to choose the action $1, 2, 3, 4$ uniformly. Thus, with probability $p$, the agent will choose the action follow the current policy while with probability $1 - p$, the agent will choose a random action. By setting different $p$, we show in Fig. 7 that the approximation error in the Actor update may significantly degrade the learning performance. Meanwhile, Fig. 7 also indicates that decreasing bias can be helpful to improve the learning performance (see the red and green lines in Fig. 7). This observation also verifies our results in Theorem 1.

## L    OFF-POLICY A-C ALGORITHEM AS NEWTON'S METHOD WITH PERTURBATION

We note that the actor and critic updates in Eqn. (9) and Eqn. (8) are a general template that admits both off- and on-policy method. More specifically, denote the target policy by $\pi_{\text{tar}}$ and the behavior policy by $\pi_{\text{bhv}}$. When the off-policy menthod is used, then the updates in Eqn. (9) and Eqn. (8) are given by

$$\omega_{t+1} \leftarrow \arg\min_{\omega} \mathbf{E}_{(s,a) \sim \rho^{\pi_{\text{bhv}}}} \left[ Q_{\omega, \pi_{\text{tar}\, t+1}}(s, a) - \omega^{\top} \phi(s, a) \right]^2,$$

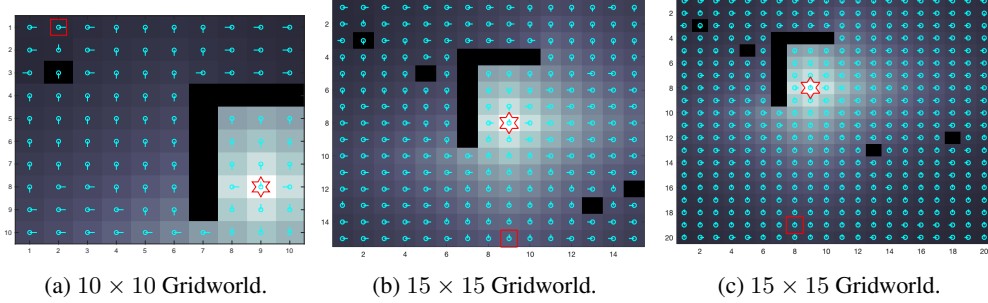

(a) $10 \times 10$ Gridworld.          (b) $15 \times 15$ Gridworld.          (c) $15 \times 15$ Gridworld.

Figure 3: Gridworld benchmark with different sizes. The colors specify the 'goodness' measure of the state, i.e., the darker color cubes are with lower $v(s)$ value and the agent should avoid those areas. The horizontal lines and vertical lines in each cube point to the direction the agent should take, i.e., policy at every state. Fig. 3(a), Fig. 3(b) and Fig. 3(c) show the learning results after 50 iterations of A-C update.

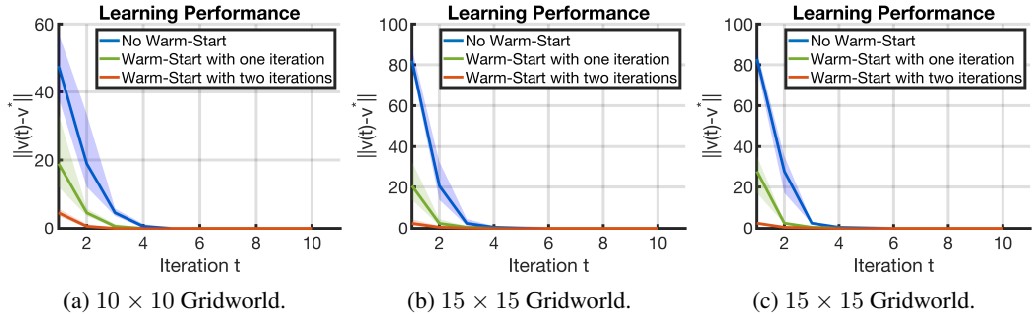

(a) $10 \times 10$ Gridworld.    (b) $15 \times 15$ Gridworld.    (c) $15 \times 15$ Gridworld.

Figure 4: The impact of the Warm-Start Policy when no approximation errors in Actor update and Critic update. The convergence behavior given different initial policy, i.e., a random policy (no Warm-Start), a Warm-Start policy obtained by running the A-C algorithm for one iteration and two iterations. The $x$-axis represents the A-C update step and $y$-axis is the value of the norm $\|\boldsymbol{v}(t) - \boldsymbol{v}^*\|$.

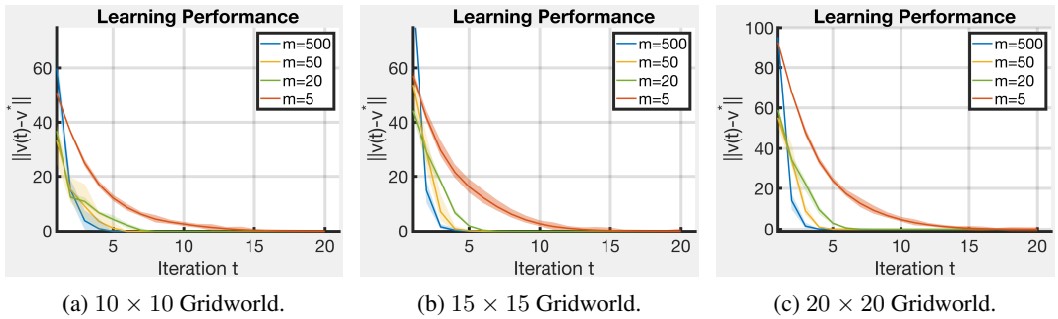

(a) $10 \times 10$ Gridworld.    (b) $15 \times 15$ Gridworld.    (c) $20 \times 20$ Gridworld.

Figure 5: Learning performance vs. rollout length.

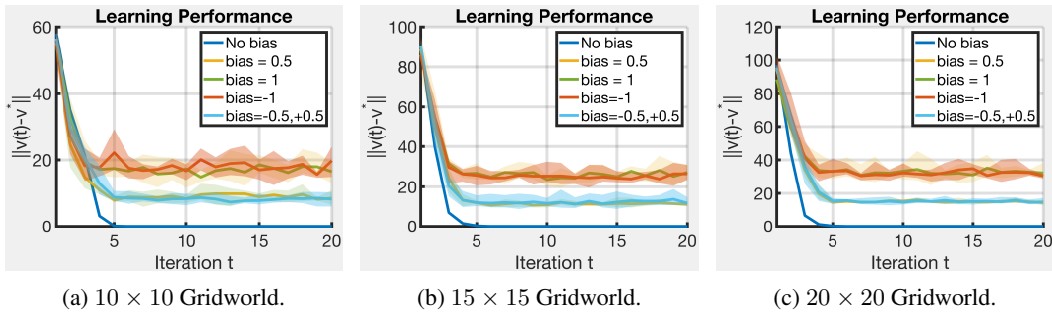

(a) $10 \times 10$ Gridworld.    (b) $15 \times 15$ Gridworld.    (c) $20 \times 20$ Gridworld.

Figure 6: Illustration of the lower bound in Theorem 1.

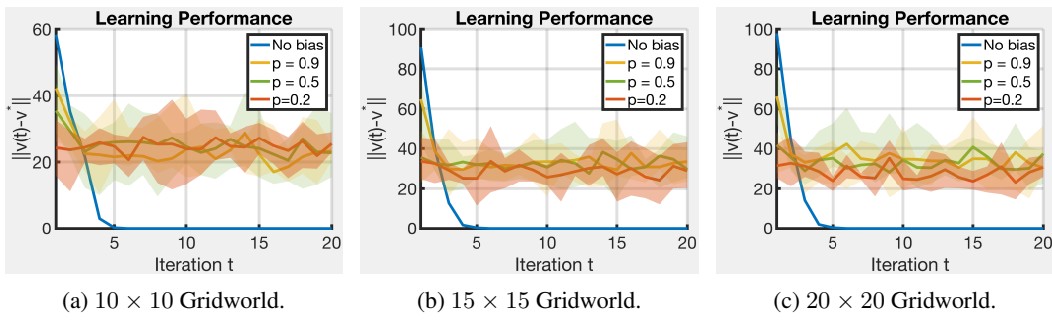

(a) $10 \times 10$ Gridworld.    (b) $15 \times 15$ Gridworld.    (c) $20 \times 20$ Gridworld.

Figure 7: Convergence behavior vs. Approximation Error in the Actor Update.

$$\pi_{t+1} \leftarrow \arg\max_{\pi} \mathbf{E}_{(s,a)\sim\rho^{\pi_{\mathrm{bhv}}}} \left[ Q_{\omega_{t+1},\pi_{\mathrm{tar},t}}(s,a) \right].$$

This is in contrast to the updates given below when the on-policy method is used:

$$\omega_{t+1} \leftarrow \arg\min_{\omega} \mathbf{E}_{(s,a)\sim\rho^{\pi_{\mathrm{tar}}}} \left[ Q_{\omega,\pi_{\mathrm{tar}\,t+1}}(s,a) - \omega^\top \phi(s,a) \right]^2,$$

$$\pi_{t+1} \leftarrow \arg\max_{\pi} \mathbf{E}_{(s,a)\sim\rho^{\pi_{\mathrm{tar}}}} \left[ Q_{\omega_{t+1},\pi_{\mathrm{tar},t}}(s,a) \right].$$

- One major challenge of the off-policy analysis lies in the fact that the behavior policy can be arbitrary Sutton et al. (1999)Sutton & Barto (2018) and hence it is impossible to develop a unifying framework. For example, the behavior policy can be obtained by human demonstration (a similar idea is used in an early version of AlphaGo), deriving from the target policy as in Q-learning/DQN or from a previous behavior policy. Meanwhile, the key drawback of off-policy method is that it does not stably interact with the function approximation and is generally of greater variance and slower convergence rate Sutton & Barto (2018). In this regard, modern off-policy deep RL requires techniques such as growing batch learning, importance sampling or ensemble method to stabilize the algorithm. Thus, for ease of exposition, we only include the on-policy analysis in our work.

- Our framework and theoretical results are able to be applied to off-policy setting with the extra assumption on the behavior policy. In particular, we assume the behavior policy is in the neighborhood of the target policy, i.e., in each Actor and Critic update step,
$$\|\mathcal{E}_{\mathrm{bhv\text{-}tar},t}\| := \|\pi_{\mathrm{tar}}, t - \pi_{\mathrm{bhv},t}\| \leq C_{bt},$$
where $C_{tb} \geq 0$ is a constant. In this way, we can write the A-C update in the off-policy setting as a Newton Method with perturbation, i.e.,
$$\boldsymbol{v}_{\pi_{\mathrm{tar}},t+1} = \boldsymbol{v}_{\pi_{\mathrm{tar}},t} - (\boldsymbol{J}_{\boldsymbol{v}_{\pi_{\mathrm{tar}},t}}^{-1}(\boldsymbol{v}_{\pi_{\mathrm{tar}},t} - T(\boldsymbol{v}_{\pi_{\mathrm{tar}},t})) - \mathcal{E}_t),$$
where $\mathcal{E}_t$ is the perturbation which captures the approximation error from Actor update, Critic update and the behavior policy. Explicitly, we have the perturbation with the following form,
$$\mathcal{E}_t = \mathcal{E}_{v,t} + \mathcal{E}_{\hat{J},t}(\boldsymbol{v}^{\hat{\pi}_{t+1}} - (\boldsymbol{r}_{\widetilde{\pi}_{t+1}} + \gamma \boldsymbol{P}_{\widetilde{\pi}_{t+1}} \boldsymbol{v}^{\hat{\pi}_{t+1}})) - \boldsymbol{J}_{\hat{\boldsymbol{v}}_t}^{-1}(\mathcal{E}_{r,t} + \mathcal{E}_{bhv-tar,t} + \gamma(\mathcal{E}_{P,t} + \mathcal{E}_{bhv-tar,t})\boldsymbol{v}^{\hat{\pi}_t}).$$
Thus, the off-policy analysis is similar to the on-policy case but with the 'error' induced by the behavior policy.

## M ONE-PAGE REVISION (SEE NEXT PAGE)

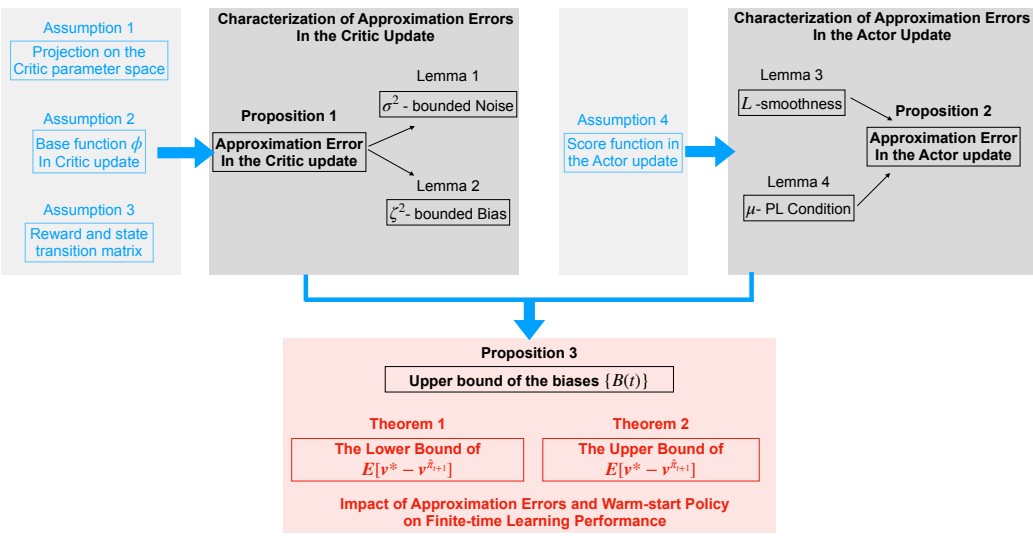

Figure 8: Illustration of the theoretical analysis.

**In Section 3.** We add the illustration of the theoretical analysis Fig. 8 to make this work more accessible to a broader audience.

**In Section 4.** We add Table 1 to summarize the key observations from our main results (Theorem 1 and Theorem 2)

| | $\mathcal{B}(t) \to 0$ | $\|\mathcal{B}(t)\| > 0$ |
|---|---|---|
| when the distance between $\pi_0$ and $\pi^*$ is small | The warm-start can facilitate the online convergence (Theorem 2) (Empirical studies:Silver et al. (2017; 2018)) | Biases can throttle the convergence significantly due to the accumulation effect (Theorem 1) (Empirical studies: Uchendu et al. (2022)) |
| when the distance between $\pi_0$ and $\pi^*$ is relatively large | The imperfections of the warm-start can be "washed out" by online learning (Theorem 2, Eqn. (30)) (Empirical studies: Bertsekas (2022b)) | The warm-start policy goes hand-by-hand with the approximation error to influence the learning performance |

Table 1: The learning performance given different warm-start policy and biases setting.In Theorem 1, we point out that the bias terms have direct impact on "*whether the warm-start policy is able to facilitate the online learning*". For instance, "even when the warm-start policy is nearly-optimal", there is still no guarantee that online fine-tuning can improve the policy much if there exist biases in the approximation errors in online Actor and Critic updates and these biases are not dealt with properly. To clarify further, consider the case when the bias is always positive, i.e., $\mathcal{B}(t) > 0$ for all $t \geq 0$, the lower bound is always positive and bounded away from zero. In Theorem 2, we aim to answer the question: "*under what conditions online learning can be significantly accelerated by a warm-start policy?*". Consider the case when the approximation error is unbiased (such that the A-C can be viewed as the Newton's Method). Clearly, we have $\|\mathbf{E}[v^{\hat{\pi}_{t+1}} - v^*]\| \leq \left(\prod_{i=0}^{t} H_{t-i}\right) \|v^* - v^{\pi_0}\|$, which can decrease quickly as long as the warm-start policy $\pi_0$ is not far away from the optimal policy $\pi^*$ and also satisfies Assumption 5. Intuitively, this result shows that "the imperfections of the warm-start policy can be 'washed out' by the (superlinear) Newton step" and corroborates with the observation in the very recent literature (Bertsekas, 2022b.). We remark that this phenomenon has not been formalized by previous works on the A-C algorithm.

