# OpenReview forum: "The Impact of Approximation Errors on Warm-Start Reinforcement Learning: A Finite-time Analysis"
_ICLR.cc/2023/Conference — Submitted to ICLR 2023_

### Official Review · Reviewer_gGbf · 2022-10-24

**Confidence:** 2
**Correctness:** 4
**Technical Novelty And Significance:** 3
**Empirical Novelty And Significance:** Not applicable
**Recommendation:** 6

**Clarity, Quality, Novelty And Reproducibility:**

This paper is well written. The author might want to highlight the difference of Theorem 2 compared with the policy gradient analysis to better show the novelty.

**Strength And Weaknesses:**

### Strength:

- This paper is overall well-written and easy to follow
- Treating actor-critic algorithms as a perturbed Newton method is novel and provides many helpful insights

### Weakness

- The proof of Theorem 2 is similar to the proof in policy gradient methods leaving some control of approximation error. The authors is expected to highlight why this analysis is novel given the existing analysis in PG.
- The impact of warm-start is not fully addressed in the theorems. In detail, in Theorem 2 and Theorem 4, the sub-optimality $h(\omega, \theta_t^*) - h(\omega, \theta_{t-1})$ is exponentially decreasing, which is similar with conventional analysis in optimization. I wonder why the author emphasizes the term `warm start' in this paper.

**Summary Of The Paper:**

This paper studies the influence of warm-start in actor-critic algorithms. The authors provide a finite-time analysis of the impact of the approximation error and the sub-optimality of the initial policy.

**Summary Of The Review:**

I don't have any other major comments besides the comments above. Here're some specific questions.

- What's the rate of $\prod_{i=0}^t H_{t-i}$ in Theorem 5? Should we expect it to be decreasing exponentially fast w.r.t. t?
- What does the `warm start' exactly mean? Does it mean by eq. (8) we could get a closed-form solution of the critic update instead of a one-step update? If that's the case, the analysis should not be claimed as a single-time scale analysis since the critic is updated completely until the actor update.
- Again, compared with the closed-form update of the critic, can we use the two-time scale [1] update to provide more efficiency?
- In the proof of Theorem 4, why does the first inequality after `plugging Eqn. 27 into Eqn 26 holds? How do we control the $\mathcal E_t$ by $\mathcal B(t)$ and $\mathcal N(t)$?
- The authors mentioned the actor-critic algorithm could be treated as a perturbed Newton method. Is there any relationship between this understanding and the natural actor-critic algorithm where [2] provides some theoretical analysis? It would be helpful to give some brief discussion.

After all, I found this paper valuable and would suggest acceptance

[1] Wu, Yue Frank, et al. "A finite-time analysis of two time-scale actor-critic methods." Advances in Neural Information Processing Systems 33 (2020): 17617-17628.

[2] Khodadadian, Sajad, et al. "Finite sample analysis of two-time-scale natural actor-critic algorithm." IEEE Transactions on Automatic Control (2022).

---

> ### Comment · Area_Chair_ovdJ · 2022-11-24
> **Thank you! Are you satisfied by the answers?**
>
> Dear reviewer,
>
> Thanks again for your detailed review! The authors have replied back to you. Please read them carefully, and acknowledge their response. If there is still an unclear point about the paper or you do not agree with some of the responses, please let them know. We would like to have a robust discussion now.
>
> If you have any further questions from them, please ask them now. We have to make the final decision soon.
> Also as a courtesy to the authors, please acknowledge their rebuttal.
>
> Thank you,
> Area Chair

---

### Official Review · Reviewer_528u · 2022-10-24

**Confidence:** 2
**Correctness:** 3
**Technical Novelty And Significance:** 2
**Empirical Novelty And Significance:** 2
**Recommendation:** 6

**Clarity, Quality, Novelty And Reproducibility:**

The paper seems to contain somewhat new results.
All the presentations need to be improved following the sense of my points given in the previous comments.


**Strength And Weaknesses:**


The paper seems to present interesting results.
In terms of presentation, there seem exist some room for further improvements.
In section 2.2, the authors explain AC method, which is a stochastic RL method.
However, when explaining how the error propagates, the derivation process uses a perfectly deterministic manner.
Therefore, it is rather confuzing.
The same points apply to section 3.
The AC method is expressed in a deterministic manner with exact expectations.
Therefore, the title of sections need to be changed and the discussions need to be changed.

In eq (10), it is not clear how hard it is to project parameters into the parameter space. Some discussions may be needed.

In Theorem 1, it is not clear if the result is for the deterministic update in eq (8) or it is for the batch-based stochastic AC in eq (10).




**Summary Of The Paper:**

In this work, the authors take a finite-time analysis approach to quantify the impact of approximation errors on the learning performance ofWarm-Start A-C method with a given prior policy. By delving into the
intricate coupling between the updates of the Actor and the Critic, the paper first provides upper bounds on the approximation errors in both the Critic update and Actor update of online adaptation, respectively,
where the recent advances on Bernstein’s Inequality are leveraged to deal with the sample correlation therein.

**Summary Of The Review:**

The paper seems to contain interesting results.
The overall presentation needs to be improved further.
Moreover, the authors need to discuss more the advantage of the proposed analysis compared to existing ones.

---

> ### Comment · Area_Chair_ovdJ · 2022-11-24
> **Thank you! Are you satisfied by the answers?**
>
> Dear reviewer,
>
> Thanks again for your detailed review! The authors have replied back to you. Please read them carefully, and acknowledge their response. If there is still an unclear point about the paper or you do not agree with some of the responses, please let them know. We would like to have a robust discussion now.
>
> If you have any further questions from them, please ask them now. We have to make the final decision soon.
> Also as a courtesy to the authors, please acknowledge their rebuttal.
>
> Thank you,
> Area Chair

---

### Official Review · Reviewer_7KE9 · 2022-10-24

**Confidence:** 4
**Correctness:** 2
**Technical Novelty And Significance:** 2
**Empirical Novelty And Significance:** Not applicable
**Recommendation:** 3

**Clarity, Quality, Novelty And Reproducibility:**

The writing needs improvement as there are many typos. The result is not surprising as similar results are well-known in stochastic approximation literature.

**Strength And Weaknesses:**

Major Comments:

(1) The AC type algorithm the authors study is a stochastic approximation algorithm for maximizing the cumulative reward. In stochastic approximation, usually the finite-time bound consists of two terms, one is called bias (or optimization error), and the other is called variance (or statistical error) [1,2,3]. The bias captures how much away in expectation the iterates are from the desired limit, and the variance captures the noise error. It is well-known in stochastic approximation literature that the bias depends on the distance between the initial condition (e.g., the warm start policy in the case of AC) and the limit, and the variance depends on the batch sample size. While the terminology is different, It seems that the authors rediscovered this phenomenon in the special case of AC, which is not surprising.

[1] Srikant, R., & Ying, L. (2019, June). Finite-time error bounds for linear stochastic approximation andtd learning. In Conference on Learning Theory (pp. 2803-2830). PMLR.

[2] Bottou, L., Curtis, F. E., & Nocedal, J. (2018). Optimization methods for large-scale machine learning. Siam Review, 60(2), 223-311.

[3] Agarwal, A., Kakade, S. M., Lee, J. D., & Mahajan, G. (2021). On the Theory of Policy Gradient Methods: Optimality, Approximation, and Distribution Shift. J. Mach. Learn. Res., 22(98), 1-76.

(2) Some assumptions are too strong and can never be satisfied. For example, in Assumption 2, assuming the smallest eigenvalue of $E_\rho[\phi(s,a)\phi(s,a)^\top]$ is bounded below by $\sigma^*$ is only possible for a specific $\rho$ but not for all $\rho$. Consider a $\rho$ with an entry that is close to one and all other entries close to zero, the corresponding smallest eigenvalue of $E_\rho[\phi(s,a)\phi(s,a)^\top]$ can be made arbitrarily close to zero. Also, Assumption 3 (2) does not hold if deterministic policies are considered.

Minor Comments:

(1) Section 2 paragraph 1: I don't think the analysis allows for $\gamma=1$. Is this a typo?

(2) The paragraph after Eq. (4), saying that "F can be viewed as the gradient of an unknown function." is not correct. Since the second derivative of $F$ (provided it is twice differentiable) may not be symmetric, F in general cannot be viewed as a gradient. It is well-known that value-based algorithms such as Q-learning are not stochastic gradient descent/ascent algorithms.

(3) Eq. (5) is not clear. I need to go back and read (Grand-Cle ́ment, 2021) to understand this equation. Probably more details are needed.

(4) Eq. (8) is confusing. The parameter $w_{t+1}$ is on both left and right of the equation.

(5) In the statement of Theorem 1, on the left-hand side, since $Q$ is a vector, I believe the authors should use a suitable norm instead of an absolute value.

(6) In the paragraph after Theorem 1, more steps of rollout can only partially decrease the error. In order to make the error arbitrarily small, one has to increase the sample size. Therefore, the "either increase N or increase m" statement is scientifically inaccurate.

(7) In Theorem 2, why is there an expectation in a high probability bound? Also, where is $p$ in the bound?

**Summary Of The Paper:**

This paper provides a theoretical understanding to the advantage of using warm-start in Actor-Critic (AC) type RL algorithms. The authors achieve this by establishing finite-time bounds of warm-start AC algorithms, which explicitly capture the impact of using warm-start. In terms of technical approach, the authors cast the AC algorithm as a Newton’s method with perturbation and bound the critic error and the actor error separately.

**Summary Of The Review:**

While the authors use different terminology, the paper essentially shows that the initial condition shows up in the bias term (or optimization error term) in AC algorithm, which is a stochastic approximation algorithm, and hence using warn start can reduce that bias term. This does not seem surprising to me as similar results were well-established in stochastic approximation literature, with applications in SGD, RL algorithms such as TD-learning, and policy-gradient type algorithms. In addition, some assumptions are too strong. In view of these two points, I feel this paper in its current form is not publishable at ICLR.

---

> ### Comment · Area_Chair_ovdJ · 2022-11-24
> **Thank you! Are you satisfied by the answers?**
>
> Dear reviewer,
>
> Thanks again for your detailed review! The authors have replied back to you. Please read them carefully, and acknowledge their response. If there is still an unclear point about the paper or you do not agree with some of the responses, please let them know. We would like to have a robust discussion now.
>
> If you have any further questions from them, please ask them now. We have to make the final decision soon.
> Also as a courtesy to the authors, please acknowledge their rebuttal.
>
> Thank you,
> Area Chair

---

### Official Review · Reviewer_uKPa · 2022-10-30

**Confidence:** 2
**Correctness:** 4
**Technical Novelty And Significance:** 3
**Empirical Novelty And Significance:** Not applicable
**Recommendation:** 5

**Clarity, Quality, Novelty And Reproducibility:**

I am not familiar with the related literature, so I cannot judge the novelty of this paper. The presentation is rather technical, and it does not make the paper easily accessible from researchers outside the narrow field of error propagation in A-C methods. To the best of my judgement, the results look reasonable. However, I did not check the proofs, which makes it hard to evaluate reproducibility and correctness.

**Strength And Weaknesses:**

After Discussion

I want to thank again the authors for their clarifications and remarkable effort in improving the paper following reviewers' suggestions. Unfortunately, the discussion revealed some important concern over the result in Theorem 2: The upper bound still includes random variables, which means that convergence rate and sample complexity cannot be straightforwardly derived. Thus, I am updating my score to a slightly negative evaluation. However, I still believe that this paper will be a nice contribution with an additional effort to make Theorem 2 more informative.

----


*Strengths*
- Providing a deeper understanding of the properties of A-C methods is paramount, given that it is widely used in practice;
- The analysis consider the dependence between samples drawn from the MDP instead of the common i.i.d. assumption;
- The main result directly connects the error with the value of the initial policy.

*Weaknesses*
- The paper is rather technical, tweaking the presentation to make it accessible to a broader audience could benefit its impact;
- The warm-start setting does not seem to be crucial for the analysis;
- The analysis consider expectations estimated through samples averages, but it does not specify how the samples are collected.

*Comments*

(Warm-start setting) From the title, abstract, and introduction, the reader is inclined to believe that the warm-start setting will play a crucial role in the analysis. Instead, from my understanding, the warm-start policy only affects the results through the value $v^{\pi_0}$ in the Theorem 5. It is not clear to me whether it is essential for $\pi_0$ to be warm-started, or the result would hold for a randomly initialized policy as well. Are the properties of the warm-start policy crucial for the analysis? E.g., does it matter that the policy is coming from its own optimization process, or perhaps that it displays good coverage properties as in (Xie et al., 2021)?

(Samples averages and Markovian samples) The paper claims to introduce refined results that account for the dependence between the samples (called Markovian samples) rather than assuming i.i.d. samples, which is unattainable in practice without a generative model. However, it is not clear to me how the samples are actually collected. Do they come from a batch of episodes or a single long trajectory? I guess the former setting would further complicate the analysis, inducing additional bias and noise coming from the sampling procedure. For the latter setting instead: Why the divergence between the initial state distribution and the stationary distribution does not appear in the bounds (see Theorem 12 in Fan, Jiang, Sun (2021))? Are we assuming that the samples are directly coming from the stationary distribution here, so that we can neglect mixing considerations?

**Summary Of The Paper:**

This paper provides a finite-time analysis of Actor-Critic (A-C) methods for reinforcement learning. First, the paper provides an upper bound to the error of the critic update due to function approximation and sample estimates. Then, it provides an upper bound to the error of the actor update due to approximating the greedy step with several steps of policy gradient, and due to the bias and noise coming from the approximation error of the critic update. Finally, the paper studies the error propagation through subsequent iterations of the A-C method, providing both a lower bound and an upper bound that relates the error to the value of the initial (warm-start) policy and an additive bias term.

**Summary Of The Review:**

Since I am not familiar with the previous works in the finite-time analysis of A-C methods, it is hard for me to judge whether the contribution is substantial. Nevertheless, I followed the main steps of the error propagation argument, and both the assumptions and the obtained upper bounds seem pretty reasonable to me. Speculating on the novelty of the Markovian samples analysis and the explicit dependence of the final bound with the value of the initial policy, I am leaning towards a positive evaluation for this paper, but I provide a borderline score that reflects my limited confidence rather than clear weaknesses in the paper.

---

> ### Comment · Area_Chair_ovdJ · 2022-11-24
> **Thank you! Are you satisfied by the answers?**
>
> Dear reviewer,
>
> Thanks again for your detailed review! The authors have replied back to you. Please read them carefully, and acknowledge their response. If there is still an unclear point about the paper or you do not agree with some of the responses, please let them know. We would like to have a robust discussion now.
>
> If you have any further questions from them, please ask them now. We have to make the final decision soon.
> Also as a courtesy to the authors, please acknowledge their rebuttal.
>
> Thank you,
> Area Chair

---

### Decision · Program_Chairs · 2023-01-20

**Decision:**

Reject

**Justification For Why Not Higher Score:**

There are technical issues with the paper, so it cannot be accepted.

**Justification For Why Not Lower Score:**

N/A

**Metareview: Summary, Strengths And Weaknesses:**

The paper theoretically analyzes the actor-critic algorithms, and specifically studies the effect of initial policy (i.e., warm-start) on the convergence. The results include Proposition 1, which upper bounds the error of the critic, Proposition 2, which upper bounds the error of the actor, and Theorems 1 and 2, which are error propagation results, providing lower and upper bounds on the value of the obtained policy compared to the optimal one depending on the error encountered throughout iterations. One key aspect of the paper is that the statistical analysis (Proposition 1) is done under the Markov samples, instead of i.i.d. samples.

The reviewers have brought up several issues. The authors have been engaged during the discussion period and tried to address most of the concerns. There are, however, some remaining issues.
- Theorem 2 is not very insightful.
First of all, the upper bound in Theorem 2 has random quantities in its RHS: $B(i)$ and $H_i$ are random. This is not a reasonable form for an upper bound. This is raised by Reviewer 7KE9Z.

Furthermore, the behaviour of terms in the RHS is not clearly specified (Reviewer gGbf).
I suppose the authors could use Propositions 1 and 2 to provide an upper bound on B(i) terms, but they have not done that.

In addition, it is not clear how H_i terms behave. An upper bound for them is provided as $M L_J ||v^{\hat{\pi}_i} - v^* ||^q$, but $v^{\hat{\pi}_i} - v^*$ (a) is random itself, and (b) is the quantity that we want to upper bound anyway.

I consider this a major issue. This theorem requires some more work before being insightful or even meaningful.

- Assumption 2 is deemed too strong. The authors provided a relaxed one during the discussion period (but after the paper revision period). Even though the new assumption is still strong, Reviewer 7KE9Z who originally opposed it does not have much issue with it anymore (private discussions). So, I do not consider this a major issue for this paper.

- Some reviewers (for example, Reviewer uKPa) mentioned that the paper is rather technical and is not accessible to the broader RL audience. I recommend the authors to consider this in their revisions.

In addition to these concerns raised by reviewers, I have the following issues:

- There is a large literature on the error propagation analysis of approximate policy iteration. This is relevant to Theorems 1 and 2 of this paper. The paper does not cite and compare with them. Some examples:

Munos, "Error Bounds for Approximate Policy Iteration," ICML, 2003.

Although Munos' Theorem 2 is stated by taking the $\limsup$, so the effect of initial policy is not seen, it is easy to see from their Lemma 4 that $||V^* - V^{\pi_k} ||_\infty \leq 2 \gamma (1 - \gamma^k)/(1-\gamma)^2 \epsilon_\text{value} + \gamma^{k-1} ||V^* - V^{\pi_0}||_\infty$,
where $\epsilon_value$ is an upper bound on the critic. This shows both the error of the critic and the effect of warm-start. It does not show the effect of error in actor though.

The actor-critic setup, where there is a possibility of making an error in the actor, is also related to classification-based approximate policy iteration literature. For instance:

Lazaric et al., "Analysis of Classification-based Policy Iteration Algorithms," JMLR, 2016. [See Eq. (8)]

Farahmand et al., "Classification-based Approximate Policy Iteration," IEEE TAC, 2015.

Even though the exact form of the results of this paper might be different from the one in the API literature, without a proper comparison, the novelty of Theorems 1 and 2 is questionable.

- I believe there is a mistake in the proof of Proposition 1.

Just before Eq. (20) on page 19, it is assumed that $|| \hat{\Phi} ||_2$ is smaller than $2/\sigma^*$. This statement is true for $\Phi$ by assumption, but it should be shown for $\hat{\Phi}$.

The proof of this proposition seems to follow Fu et al 2020's Lemma 5.1. Looking at that result, we see that this step is handled by a matrix Bernstein inequality to show that $|| \hat{\Phi} - \Phi ||_2$ is small. That paper uses Tropp, 2015 for the matrix Bernstein inequality.

It appears that this latter reference does not handle Markov samples. As a result, that step has not been done. I realize this is merely a technical difficulty, but so is the analysis of Markovian samples in this work.

Another issue in the same proof is the upper bounding $P( ||Z||\geq t )$, where the Bernstein inequality is used. I suppose f in Theorem 3 corresponds to the norm of Z. But is the expected  norm of Z equal to zero as required by Theorem 3? (I don't think so.)

Overall, I believe this paper is a good attempt at analyzing actor-critic algorithms and might have some novel contributions in it, but it requires some more work before being ready for publications. Some results are not properly presented, some results may not be correct, and some others may not be as novel as portrayed. I encourage the authors to revise their work and resubmit to a future venue.